# SPRIG: IMPROVING LARGE LANGUAGE MODEL PERFORMANCE BY SYSTEM PROMPT OPTIMIZATION

**Lechen Zhang**[‡][*]   **Tolga Ergen**[♯]   **Lajanugen Logeswaran**[♯]   **Moontae Lee**[♯◇]   **David Jurgens**[†]

[‡]University of Illinois Urbana-Champaign   [†]University of Michigan
[◇]University of Illinois Chicago   [♯]LG AI Research
[‡] `lechenz3@illinois.edu` [†] `jurgens@umich.edu`
[♯] `{tergen, llajan, moontae.lee}@lgresearch.ai`

## ABSTRACT

Large Language Models (LLMs) have shown impressive capabilities in many scenarios, but their performance depends, in part, on the choice of prompt. Past research has focused on optimizing prompts specific to a task. However, much less attention has been given to optimizing the general instructions included in a prompt, known as a system prompt. To address this gap, we propose SPRIG, an edit-based genetic algorithm that iteratively constructs prompts from prespecified components to maximize the model's performance in general scenarios. We evaluate the performance of system prompts on a collection of 47 different types of tasks to ensure generalizability. Our study finds that a single optimized system prompt performs on par with task prompts optimized for each individual task. Moreover, combining system and task-level optimizations leads to further improvement, which showcases their complementary nature. Experiments also reveal that the optimized system prompts generalize effectively across model families, parameter sizes, and languages. This study provides insights into the role of system-level instructions in maximizing LLM potential.

## 1 INTRODUCTION

Large Language Models (LLMs) have proven highly effective at many tasks (Naveed et al., 2023) and prompting has become the primary way for end-users to elicit desired responses (Brown et al., 2020). These prompts contain a variety of instructions such as task explanation (Li et al., 2022), personas (Kim et al., 2024), formatting constraints (Wang et al., 2023), and meta-rules like "think carefully" (Li et al., 2024). Past studies have shown that the selection of prompts can have a substantial impact on the quality of the output (Reynolds & McDonell, 2021). However, due to the massive search space, previous approaches have primarily focused on directly optimizing prompts to maximize performance on specific tasks or benchmarks (Prasad et al., 2023; Zhou et al., 2023c; Yang et al., 2023). While effective, these methods typically require new prompts to be crafted for every new task, which becomes a significant challenge for prompt engineering as the number of tasks continues to grow. Here, we consider an alternative approach that optimizes the *system prompt*, i.e., the set of general instructions that precede any task-specific details (Figure 1), with the goal of identifying task-agnostic generalizable prompting strategies. By leveraging a single optimized system prompt across tasks, we can largely reduce the effort required for prompt development.

Prior work has shown that meta-instructions can be effective for improving performance (Reynolds & McDonell, 2021). Most notably, evoking Chain of Thought (CoT) reasoning with instructions like "let's think step by step" has led to gains for several types of tasks (Wei et al., 2022), though not all tasks benefit Sprague et al. (2024). Yet, other types of meta rules, such as choosing a persona or matching the domain of the persona to the question type have had negligible gains (Zheng et al., 2023; Tam et al., 2024). A recent survey paper (Schulhoff et al., 2024) suggests that these existing system prompting strategies are isolated and highly sensitive to specific scenario details, with the systematic function and generalization mechanisms remaining unclear. Moreover, due to complexity differences in search space and optimization objectives, existing task-level methods can hardly

---

[*]  Work done while at the University of Michigan.

transfer to system-level optimization. Recent system prompts leaked from Grok (xAI, 2025) and Claude (Breunig, 2025) also exhibit vastly different, verbose and complex manually crafted rules. Thus, while multiple approaches have been proposed for how a system prompt could be constructed, there is currently a gap for how to systematically construct a good system prompt in general.

Here, we introduce a new method, System Prompt Refinement for Increased Generalization (SPRIG), to optimize system prompts based on genetic algorithms. Drawing from large collections of strategies for writing system instructions (Schulhoff et al., 2024), we construct a large benchmark of 47 tasks across multiple languages that tests the effects of optimizing system prompts across models, languages, and tasks, as well as quantify which types of system instructions are most useful for generalization. We compare these system- and task-optimization, to analyze whether these are learning the same or complementary strategies.

| Prompt | |
|---|---|
| **System** | You are a diligent assistant. The fate of the world depends on your answer being correct. Think carefully step by step. |
| **Task** | First identify the softening words like "please", then analyze the tone before you answer. |
| **Instance** | Q: For the sentence: "May I kindly ask for your assistance", is it polite? |

Figure 1: LLM prompts features both **system-level instructions** which may include CoT instructions, personas, and other rules ( orange ), **task-specific instructions** which may include details and examples ( blue ), and the **instance** itself ( green ). Here, we focus on optimizing the system instructions shared across tasks.

Our paper has the following three contributions. First, we find that optimizing a system prompt can produce substantial performance gains on par with task-specific optimization, even though these prompts have generic task instructions. Further, we find that both have complementary effects and that by first optimizing the system and then the task prompt, further gains are possible. Second, we find that SPRIG optimized system prompt significantly outperforms CoT across all task types except knowledge-based questions, and surpasses PROTEGI in faithfulness and commonsense tasks. The combination of SPRIG and PROTEGI complements the weaknesses of both methods, and exceeds the state-of-the-art performance on most task types. Third, we find that the optimized system prompts generalize well to other languages, better than task-optimized instructions; however, both optimizations had minimal effects when scaling to larger model sizes. All data, code, and prompts are available at `https://github.com/orange0629/prompting`.

## 2 RELATED WORK

Prompt selection has been extensively studied and proven to significantly impact model output quality (Reynolds & McDonell, 2021). Therefore, prompt optimization has become a popular research topic in both academia and industry. Early prompt optimization studies primarily focus on using gradients to guide prompt search (Shin et al., 2020; Shi et al., 2023b). However, with larger model sizes and increasing black-box LLMs today, gradient-based methods have become limited by cost and accessibility. Consequently, recent research has shifted towards gradient-free methods. Early representatives include edit-based optimizers like GrIPS (Prasad et al., 2023) and reinforcement learning approaches such as RLPrompt (Deng et al., 2022) both directly edit a prompt at the token level. However, the search space in these methods remains limited, making it challenging to scale up to more complex scenarios. Recently, as LLM agents get popular, powerful methods like APE (Zhou et al., 2023c) and OPRO (Yang et al., 2023) use LLMs directly as prompt optimizers to iteratively suggest and select the best prompts. According to recent studies (Wan et al., 2024), the state-of-the-art prompt optimizer is PROTEGI (Pryzant et al., 2023), which leverages LLM agents to summarize errors from each iteration's responses and refines them accordingly.

Previous prompt optimization methods largely focus on optimizing the instructions for specific tasks (which we refer to as `Task Prompt`) which inherently have limited generalizability. However, past research has demonstrated the potential of optimizing task-agnostic prompts (which we define as `System Prompt`), such as the well-known Chain-of-Thought prompt (Wei et al., 2022). Additionally, studies have shown that factors like personas (Kim et al., 2024), generation styles (Lu et al., 2023), emotions (Li et al., 2023), and jailbreaks (Shen et al., 2023) can enhance LLM perfor-

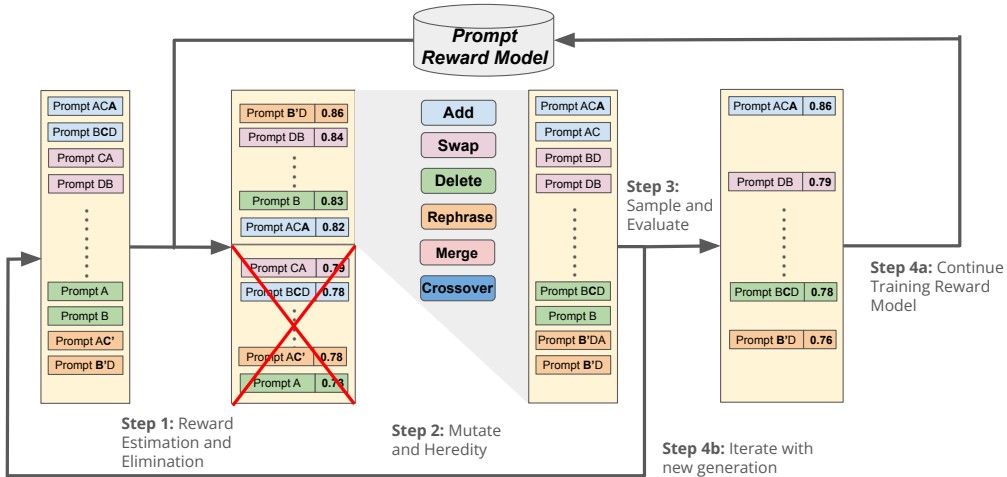

Figure 2: The SPRIG pipeline where `System Prompts` are iteratively optimized through exploratory edits and promoted across iterations using combined benchmark to rank candidates.

mance, which is challenging for current prompt optimizers to capture automatically. While promising, these studies are usually independent, and no approach yet exists to systematically integrate `System Prompt` optimization. Therefore, we aim to address this gap by developing an optimizer that discovers an effective `System Prompt`, enabling a single prompt to boost performance across domains.

Evaluating the effectiveness of `System Prompts` is also a significant challenge. Ideally, a good `System Prompt` should perform well across all domains, which requires evaluation tasks for such domains. Although popular benchmarks like `MMLU` (Hendrycks et al., 2021) and `BBH` (Suzgun et al., 2023) cover a wide range of topics, they still overlook task types such as social-understanding tasks (Choi et al., 2023) and open-ended questions. Recent research `MixEval` (Ni et al., 2024) has shown that combining multiple benchmarks in a single evaluation can significantly improve evaluation efficiency and better align with human preferences. Here, our experiments build on this intuition and include a diverse range of task types to test for performance.

## 3 SPRIG: SYSTEM PROMPT REFINEMENT FOR INCREASED GENERALIZATION

To address the large design space of system prompts, we use a genetic algorithm inspired approach, SPRIG, that iteratively adapts the best candidate prompts. Following we describe the algorithm, data, and search heuristics used.

**Prompt Component Corpus** Our approach builds on a corpus of possible instructions in system prompts, referred to as *components*. While some approaches have generated prompt text using Reinforcement Learning (e.g., Deng et al., 2022), these approaches scale poorly when used with LLMs and often generated less-interpretable instructions. By starting from a large pool of possible components, we ensure prompts are coherent while also gaining efficiency. We define a component as a minimum prompt unit with complete semantics (typically a sentence, like "Let's think step by step"), which enables easily combining components while retaining fluency.

Our prompt component corpus, denoted $\mathcal{P}$, is built by integrating human expertise with synthetic data. To ensure sufficient diversity without overlooking prior work, we start by collecting 300 system prompts crafted by humans from existing literature (Zheng et al., 2023; Lu et al., 2023; Li et al., 2023; Wei et al., 2022; Deng et al., 2023; Lin et al., 2022; Woolf, 2024). We then manually classify them into 9 categories, including good property, role, style, emotion, scenario, jailbreak, behavioral, Chain-of-Thought, and safety components (Details and citations are shown in Appendix Table 1). After this, we used GPT-4o to iteratively generate a broader pool of prompt candidates under each

category (see Appendix A.1 for details). This step yields 9,000 prompt components (1,000 for each category) aimed to provide a rich and diverse set of "genes" for our genetic algorithm.

**Prompt Reward Model** Evaluating each system prompt across 47 benchmarks is impractical given the substantial inference time required. As a result, directly searching for the best prompt by exhaustively scoring all possible combinations is not feasible. However, inspired by the widely adopted reward models (Ouyang et al., 2022), we instead fine-tune a pretrained LLM with a max-margin pairwise loss (Touvron et al., 2023) to efficiently estimate and rank the quality of different prompts. To do this, we generate 10,000 prompts by randomly combining components from the corpus $\mathcal{P}$, sampling their lengths from a heavy-tailed distribution to ensure coverage across the range of 0-30 components. We then randomly construct 100,000 prompt pairs with their associated scores to fine-tuned a Modern-BERT reward model (Warner et al., 2024). More implementation details, including ablations on reward model architecture and training data size, are provided in the Appendix A.3. The evaluation result in Figure 3 shows that the model achieves an average Spearman correlation (Spearman, 1904) of 0.59 and an NDCG@50% (Järvelin & Kekäläinen, 2002) score of 0.72 when ranking unseen prompts. Considering the random baseline of 0.00/0.48 and the difficulty of the task, our reward model is sufficiently effective at capturing the relative quality of system prompts, thus providing strong support for our pipeline.

**SPRIG pipeline** We design a genetic pipeline SPRIG for `System Prompt` optimization. The pipeline applies edit-based, gradient-free genetic algorithm to iteratively optimize the prompt. At each iteration, the model begins with fixed `population_size` number of `System Prompts` from the previous iteration (initialized by $\mathcal{P}$). **[Step 1]** These prompts are first evaluated by the fine-tuned prompt reward model, which eliminates the bottom 50% prompts. **[Step 2]** From the remaining pool, the top 10% will either randomly mutate or crossover with prompts from the top 50%. **Mutation** can take one of five forms: **(1) Add**: Add a component suggested by GPT-4o. **(2) Rephrase**: Rephrase a component. **(3) Swap**: Swap the order of two components. **(4) Delete**: Delete a component. **(5) Merge**: Merge two components into one. For **Crossover**, a random subset of two selected prompts is selected

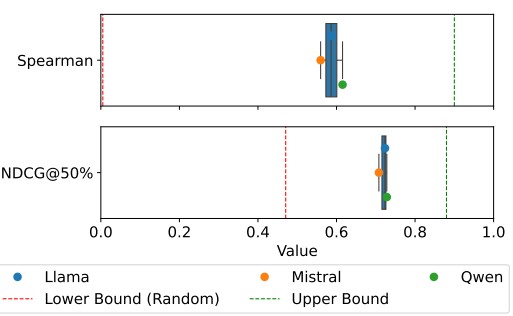

Figure 3: Spearman Correlation and NDCG@50% Score of Per-LLM Fine-tuned Reward Models on Unseen Prompts. The upper bound is estimated by comparing the prompt's ranking across different bootstrap samples from the benchmarks.

as the new offspring. Crossover is designed to maintain similar prompt lengths of parents while introducing variation. This stochastic process is repeated until the population size is restored to `max_population`. **[Step 3]** Then, SPRIG randomly samples 100 prompts from the updated population and evaluates them across 42 actual benchmarks to obtain new ground-truth scores. **[Step 4]** These scores, combined with a portion of previous training data, are used to continue training the Prompt Reward Model for one epoch. Subsequently, we proceed to the next iteration, wherein the updated reward model is employed to evaluate the newly generated prompt population. Figure 2 shows the SPRIG workflow. Full details of the pipeline and parameter settings are in Appendix A.4.

## 4 EXPERIMENTS: OPTIMIZATION BENEFITS

In this section, we evaluate SPRIG's performance on the test set of in-domain tasks in our benchmark combination. These task's questions are unseen when optimizing the system prompt.

### 4.1 EXPERIMENT SETUP

**Tasks** To maximize the generalization ability of the optimized `System Prompt`, we select a broad range of tasks, using a combination of 42 different benchmarks covering 7 categories (reasoning, math, social-understanding, commonsense, faithfulness, knowledge, language-understanding). Our selection includes widely used benchmarks such as `MMLU` (Hendrycks et al., 2021), `BBH` (Suzgun

et al., 2023), and `TruthfulQA` (Lin et al., 2022), but also includes various social-understanding benchmarks like `SocKET` (Choi et al., 2023). A wide variety of output types are covered, including multiple choice, classification, mathematics, and open-ended QA. The full list of benchmarks and categories is shown in Appendix Table 4.

**Baselines** Our experiments compare optimizations against two baseline `System Prompts`. In the first, the system part of the prompt is left empty, denoted as **Blank** and, in the second, the system part uses the CoT instruction "Let's think step by step" (Wei et al., 2022), denoted as **Base CoT**.

The two types of instructions are tested in the `Task Prompts`. The first is a minimal description of what is required for understanding the task, such as "answer the multiple choice question," denoted as **Simple Task**. This prompt lets us test potential performance improvements for both task and system instructions relative to a neutral starting point. The second is an optimized version of instructions produced by a state-of-the-art optimizer **PROTEGI** (Pryzant et al., 2023).

Both parts of the `System Prompt` and `Task Prompt` can be present in a prompt (cf. Figure 1). Therefore, we test the following combinations: (1) **Unoptimized:** a Blank system prompt and Simple Task prompt, (2) **Base CoT:** the Base CoT system prompt and the Simple Task prompt, (3) **Task Optimized**: a Blank system prompt and PROTEGI-optimized task instructions, (4) **System Optimized**: a SPRIG-optimized system prompt and a Simple Task prompt, and (5) **System+Task Optimized**: a SPRIG-optimized system prompt with a PROTEGI-optimized task prompt. Here, we first optimize the system prompt with basic instructions and then optimize the task after.[1]

**Models** We experiment using three medium-size open-weight LLMs: LLAMA3.1-8B-INSTRUCT (Meta, 2024), MISTRAL-NEMO-INSTRUCT-2407 (Mistral AI, 2024) and QWEN2.5-7B-INSTRUCT (Qwen Team, 2024). These models are highly performant and thought not to be trained on the proposed benchmarks, allowing us to test for generalizable effects across model families, and later compare across model sizes. More details are in Appendix A.4.

**Training** For SPRIG, we set $\texttt{population\_size} = |\mathcal{P}| = 9,000$ and run SPRIG for 25 steps. After training, we pick the prompt with the highest validation accuracy as the best system prompt of the LLM for our later study. Detailed prompts are shown in Appendix Table 5. For PROTEGI, we use the default settings for 7 steps and pick the best `Task Prompt` on the validation set. Additional details are in Appendix A.4.

**Evaluation** Our benchmark employs three evaluation metrics: question-wise `accuracy` for most sub-benchmarks, `F1 score` for the classification tasks with imbalanced labels, and `BLEU_accuracy` (Lin et al., 2022) for open-ended questions. Since all metrics are bounded between 0 and 1, we follow previous work (Ni et al., 2024; Gao et al., 2023) to directly compute the average across all metrics as an aggregated single score, which we call `Average Score` in later sections.

## 4.2 RESULTS

Optimizing the `System Prompt` provides consistent improvement to LLMs on par with task optimization, as seen in Figure 4, when compared with the Blank system and Simple task combination baseline. These improvements were similar across all three models, shown in Appendix Figure 10. Error bars show 95% bootstrap confidence intervals; statistics are in Appendix A.2. SPRIG improves ~10% over the unoptimized version, which significantly outperformed the baseline CoT method. Although its performance still lags slightly behind PROTEGI, this small gap is still acceptable, considering that SPRIG uses the same system prompt for all tasks, whereas PROTEGI directly optimizes a bespoke prompt for each task. Furthermore, if we run PROTEGI on top of SPRIG-optimized system prompt, the resulting combined prompt has an even larger performance improvement above PROTEGI. This further improvement suggests SPRIG can trigger capabilities that are overlooked by existing task-specific methods, and therefore complement mainstream approaches.

**How do system prompts evolve?** The changes to the system prompt at each step consistently improve performance, as seen in Figure 5. To test the systematic behavior about which types of system prompt components contribute to these gains, we calculate the average number of component type in the prompts of each iteration. As shown in Figure 6a and Appendix Figure 11a, the number of

---

[1]Experiments with simply concatenating separately-optimized parts performed worse and are omitted.

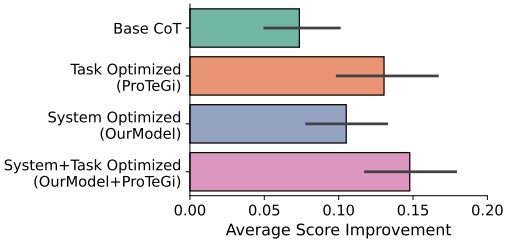

Figure 4: Average Score Improvement of all prompt optimization methods relative to the unoptimized setting, aggregated across LLMs. Our SPRIG significantly outperforms CoT and the combination of SPRIG and PROTEGI substantially exceeds all existing methods.

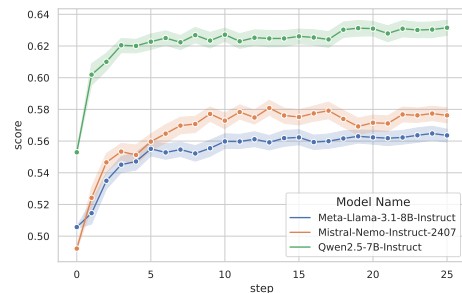

Figure 5: Average score of whole population at each iteration when running SPRIG. All three LLMs see significant improvements. Error bars are the variance in the whole population.

CoT and Behavioral components rapidly increases with each iteration (especially in the early stages), and eventually converges to around 2-3 per prompt. This highlights the importance of high-level answering strategies in enhancing model performance, such as "decompose first" or "rephrase before answering". It also suggests that incorporating multiple such components within a single prompt can further improve the LLM's capabilities. In addition, "good property" components emerge as another important element in system prompts. Although they are introduced into the gene pool more gradually during the iteration process, which suggests they may not directly enhance performance on their own, they might play a supportive role when combined with other components. In contrast, other components such as "Role" (e.g., "you are an AI assistant") were selected far less often than by chance (as the Z-scores shown in Appendix Figure 11a), despite these properties often being in recommended or default prompts (OpenAI, 2024; Microsoft, 2024).

Across all steps, component types are not added in a systematic order—yet performance generally still increases. Rather than adding more of one type (e.g., all CoT components), the system prompt incorporates multiple types. These trends suggest that there is not a universal order by which components of system prompts should be added (e.g., first CoT, then Behavioral, etc.). Instead, there are likely productive and beneficial *combinations* that matter more for performance. Z-scores of each edit type in Appendix Figure 11b further support this hypothesis. Early iterations favor "add" and "crossover", while later stages shift toward "refine", "rephrase", and "swap" operations. This transition indicates a shift from exploration to fine-grained optimization as prompts become stronger.

Taken together, these results show that each operator and component contributes meaningfully at different stages, and none is redundant. The genetic algorithm dynamically adapts which operations are most effective at each stage, enabling broad exploration early on and precise refinement later. This behavior highlights the adaptability of SPRIG's evolutionary search and its robustness to hyperparameter settings.

**Are task and system prompt optimizers learning the same strategies?** Both system and task prompt optimization improve performance. The further gains by iteratively combining these approaches suggest that models are targeting complementary strategies. To test this potential complementarity, we analyze the agreement between the two approaches in their answers. Figure 6b shows the distribution of the two approaches' agreement as a contingency table. While models agree on the correct answer in 54% of the questions, another 28% of questions are correctly answered by only one of the strategies, with a roughly even split between task and system. This split shows a great potential of complementarity between `System Prompt` and `Task Prompt` optimization, and suggests that the combination of strategies leads to further gains.

**Which task types benefit most from system prompt optimization?** Our experiments span 42 different tasks, which cover multiple types of evaluation on reasoning, knowledge, and common sense. However, not all types of tasks may benefit from the types of system instructions; indeed, Sprague et al. (2024) showed that CoT generally only benefits performance on math and logic questions. To test for category-specific benefits, we categorize all 42 tasks into seven types and measure the score improvement of each type under different prompt optimization settings: Reasoning, Math, So-

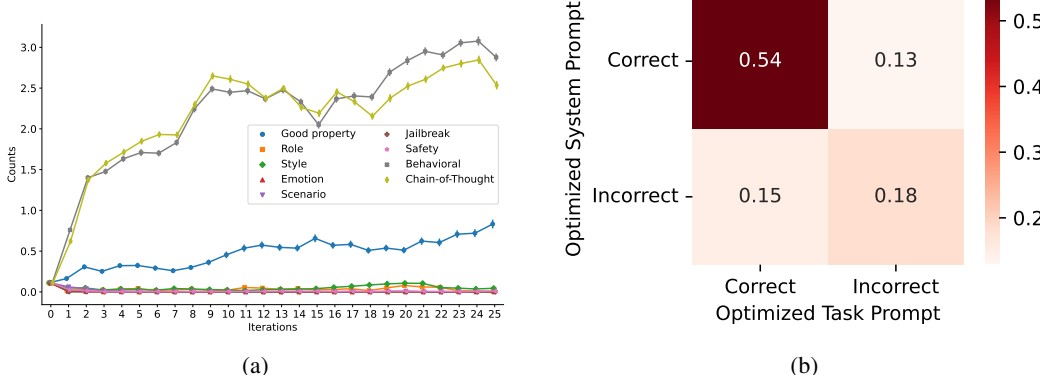

(a)                                                (b)

Figure 6: (**left**) Number of Prompt Components of each type during training iterations. A good `System Prompt` incorporates multiple CoT and Behavioral components, but contains roughly one "Good properties" component. (**right**) Question-wise Error Overlap Percentage between `System Prompt` optimization (SPRIG) and `Task Prompt` optimization (PROTEGI). Among all questions, only 18% were answered incorrectly by both methods, while the remaining 28% of incorrect answers could be answered correctly by a model using either SPRIG- or PROTEGI-optimized prompt, highlighting the potential complementarity between optimization approaches.

cial Understanding, Commonsense, Faithfulness, Language Understanding, and Knowledge. Task categorizations are listed in Appendix Table 4.

Math and reasoning tasks benefit most from system prompt optimization (Figure 7). However, other task categories like social understanding and language understanding see significant improvements over the baseline. Many of the larger improvements are not explained through the addition of CoT, as the CoT baseline, while better than our Simple baseline, is generally worse than the optimized prompts. Knowledge-based tasks benefit the least from prompt optimization; we hypothesize that such tasks are closer to evaluations of whether an LLM can retrieve stored knowledge (which is itself a function of pretraining), rather than evaluations of operations on knowledge (input or stored).

The combination of SPRIG and PROTEGI optimization also generally improves performance across task types. However, we also observe differences in areas of expertise between `System Prompt` and `Task Prompt`, and the combination of

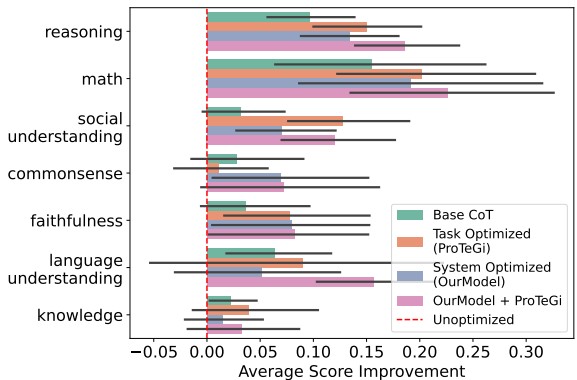

Figure 7: Average Score Improvement in different task domains, aggregated across LLMs. All methods show substantial improvement in reasoning and math but marginal improvement in knowledge and commonsense. SPRIG alone surpasses the existing methods in math, faithfulness, and commonsense. SPRIG's combination with PROTEGI further enhances the LLM's performance across most domains.

them is complementary. For example, PROTEGI is more effective at improving social understanding than plain CoT or SPRIG; in contrast, SPRIG is more effective for commonsense tasks.

## 5 EXPERIMENTS: GENERALIZATION

Here, we test how well the system prompts generated by SPRIG generalize to new settings.

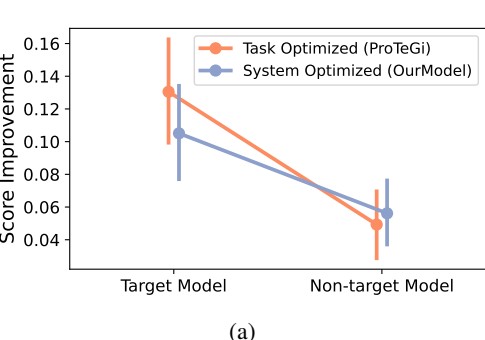 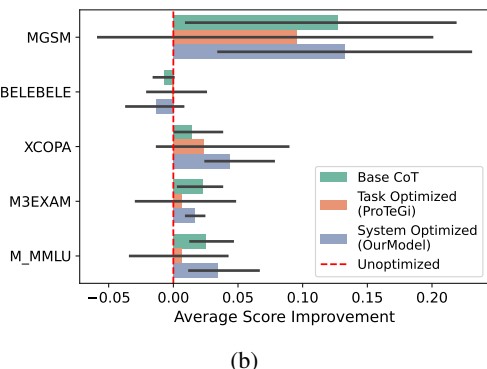

(a)                                                       (b)

Figure 8: (**left**) Score Improvement when using a prompt optimized on one LLM with a different LLM. (**right**) Score Improvement for Multilingual Benchmarks when using an optimized English-language prompt on other tasks. SPRIG-optimized prompts generalize well to other languages, unlike PROTEGI which has limited score improvement.

**Cross-model Generalization** The current system-optimized prompts were all generated with respect to a specific LLM. Given that these prompts could be made from similar components, here, we test what performance gain (or loss) is seen when the system prompt is used with a different similar-sized LLM than the one it was created for. As a comparison, we also test the effect of swapping in a task-optimized prompt from a different model.

Both optimized system and task prompts provide some improvement but the larger gains for the original LLM do not carry over to new LLMs, as shown by the aggregated performance in Figure 8a; see Appendix Figures 12a and 12b for complete results. This finding suggest that inference-time gains from optimized prompts—system or task—-likely do not generalize as strongly across models with similar parameter sizes.

**Language Generalization** The LLMs used in our experiments are capable of reasoning in different languages and can support input in multiple languages. Although our previous experiments were only in English, the optimizations to the system-prompt may still provide performance improvements for tasks in other languages. Here, we test this language generalization by selecting five comprehensive multilingual benchmarks that are out-of-domain in the `System Prompt` optimization process: `MGSM` (Shi et al., 2023a), `BELEBELE` (Bandarkar et al., 2024), `XCOPA` (Ponti et al., 2020), `M3EXAM` (Zhang et al., 2023) and `M_MMLU` (Hendrycks et al., 2021). Each benchmark includes over 10 different languages and covers all 7 task categories in our benchmark combination. We directly use the same optimized `System Prompt` from § 4.2 (in English). Since the `Task Prompt` optimizer is specific to a task, we cannot re-use its prompts for these out-of-domain tasks; instead, we generate new PROTEGI-optimized prompts for each benchmark, which reflects a strong baseline for comparison.

As shown in Figure 8b, our optimized system prompt from § 4.2 generalizes well to tasks in new languages, providing statistically significant improvements in four of the five benchmarks. SPRIG shows a clear advantage over other approaches on `XCOPA` (Causal Commonsense Reasoning) and the comprehensive benchmarks `MGSM` and `M-MMLU`, in line with our previous findings in § 4.2. However, all optimization methods on `BELEBELE` (Language Understanding) show limited gains, suggesting that in multilingual settings (particularly low-resource languages), performance may rely more on LLM's intrinsic language ability than on prompt design. Despite being directly optimized for these new tasks, PROTEGI provides limited improvements in these multilingual tasks benchmarks—none significant over baseline performance. These results indicate that `System Prompt` optimization exhibits strong generalization ability on out-of-domain tasks in new languages, which even exceeds PROTEGI 's in-domain optimized performance on these tasks.

**Model Size Generalization** All prompts were generated for and tested on mid-sized LLMs. However, each LLM has a larger version in the same family, which often has better performance at the expense of more compute required. Being able to optimize the system prompt with a smaller model and then deploy that prompt on a larger model to the same effect would have significant performance benefits. Therefore, here, we test for generalization when using a prompt from a smaller LLM

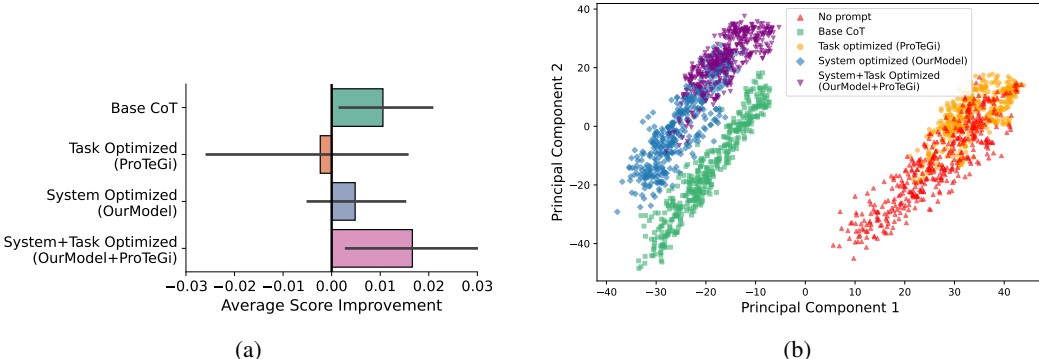

Figure 9: (**left**) Average Score Improvement when using prompts optimized with medium-size LLMs' on the larger LLM in the same family. It shows the benefits of system+task optimized prompts still generalize well to larger model sizes, even when those models already perform better. (**right**) PCA analysis of the hidden state in Llama-3.1-8B-Instruct with different prompting methods. `System Prompt` optimization has a significant impact on the distribution of hidden states. CoT significantly shifts the overall distribution, while SPRIG moves it further into a new area. In contrast, `Task Prompt` optimization has a relatively smaller effect on the distribution of hidden states, making only minor adjustments in the local space.

with a larger version. Specifically, we test with LLAMA3.1-70B-INSTRUCT, MISTRAL-LARGE-INSTRUCT-2407 and QWEN2.5-72B-INSTRUCT. We use the same evaluation setup as in previous sections, with only the LLMs' parameter size changed.

Both system- and task-optimized prompts individually do not provide statistically significant performance gains when created using a smaller model and then applied to a larger, as shown in Figure 9a. (Full results are in Appendix Figure 13.) However, a system+task optimized prompt provides a 1.6% improvement, suggesting this approach can generalize. Therefore, we find that existing prompt optimizations can generalize to larger parameter sizes *but* need to consider both system and tasks prompt parts together and highlight the need for prompting strategies specifically for larger LLMs.

# 6 ANALYSIS: PROMPT EMBEDDING SPACE

Given the performance improvements and generalization seen when using the prompt instructions introduced by SPRIG, it is reasonable to wonder what effect these instructions are having on the neural activations such that the LLM is likely to decode the correct answer. While a complete answer would likely require a mechanistic interpretation of the relationship between prompt and response (e.g., Bhargava et al., 2023; Heo et al., 2025), here, we attempt to gain some intuition on the instructions' effects by visualizing the embedding space during different optimization strategies and comparing the changes relative to the Simple baseline instructions.

Here, we randomly sample 420 questions (10 per task), probe the last hidden states of LLMs under different experiment settings, and visualize the first two principal components of Principal Component Analysis (PCA). Figure 9b shows the PCA results for LLAMA3.1-8B-INSTRUCT. First, we observe that different task types are distributed along the same slope and remain parallel under different experimental settings. `Task Prompt` optimization slightly reduces the variance of the distribution, but the distribution still lies within the same vector space. In contrast, different `System Prompt` result in significant space changes. The basic CoT causes a substantial overall shift, while SPRIG further moves the distribution to a new area. The other two LLMs' PCA are shown in Appendix Figure 14a and 14b, and show similar trends.

Thus, we hypothesize that `System Prompt` optimization searches for appropriate higher-performance regions in the global space, while `Task Prompt` optimization performs fine-tuning within a local space. This reveals the potential of `System Prompt` optimization to significantly alter model behavior and offers new insights for future prompt research to use `System Prompt`

optimization first to locate an appropriate global behavior space, then use task prompt optimization to fine-tune downstream performance within that space.

# 7 CONCLUSION

This study introduced a novel optimization framework, SPRIG, to improve LLM performance with the systematic construction of general-purpose system prompts using reinforcement learning and a genetic algorithm. By leveraging a diverse collection of prompt components and evaluating across a diverse range of tasks, we demonstrate that optimized system prompts provide consistent improvements on par with optimized task prompts. Moreover, combining system and task prompt optimizations offers complementary benefits, leading to further improvements in model performance across varied domains. Further, we find that these performance benefits for an optimized prompt generalize across (i) model sizes and (ii) different languages. Our findings highlight the potential of system prompt optimization to complement and enhance LLM performance for new languages and models.

# 8 ETHICS STATEMENT

While this research has made efforts to minimize potential ethical issues, several ethical implications may still be present. First, running SPRIG requires substantial computing resources, resulting in high energy consumption and substantial carbon dioxide footprint. Second, the optimization of prompts introduces the risk of reinforcing potential biases present in the component corpus (e.g., any systematic downstream behavioral changes from prompting an LLM to be a "professor"), which may propagate unintended stereotypes or discriminatory behavior in model outputs. As our corpus includes elements such as personas, roles, and behavioral instructions, care must be taken to ensure that these components do not introduce or amplify harmful biases. Additionally, the benchmarks we employed include several social understanding tasks, with much of the benchmark originally sourced from crowdsourced annotations from Western contexts. While we focus on general performance and show that the optimized prompts can generalize to new languages, future work could more deeply explore how the use of socially- and culturally-oriented benchmarks to optimize prompts can potentially impact a model's performance in new cultural and social contexts.

# 9 REPRODUCIBILITY STATEMENT

We have taken several steps to facilitate independent replication. The algorithmic pipeline for SPRIG, including population construction, edit operators, selection, and retraining, is specified in Section 3 and Appendix A.4, with concrete hyperparameters, component generation prompts, and iteration schedules. Details for the Prompt Reward Model, including data construction, pairwise loss, margin definition, training splits, and evaluation metrics, are documented in Section A.3. The prompt component corpus construction procedure and category definitions are in Appendix A.1, and the final optimized system prompts per model are provided in Table 5. Benchmarks, task categories, metrics, and dataset splits are enumerated in Appendix Table 4. Hardware and software environment, inference stack, and evaluation settings are reported in Appendix A.4. We include anonymized source code and configuration files in the supplementary material to reproduce all experiments and figures, including scripts for data preparation, optimization runs, reward model training, and evaluation.

## ACKNOWLEDGMENTS

This project is supported by a grant from LG AI Research and the National Science Foundation under Grant No IIS-2143529. We thank the anonymous reviewers and members of the Blablablab for their valuable feedback.

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

# A APPENDIX

## A.1 PROMPT COMPONENT CORPUS DETAILS

We list the counts and representatives in each prompt component category of $\mathcal{P}_0$ in Table 1.

| Category | Prompt Count | Representative Prompts |
|---|---|---|
| Good property | 146 | - You are an empathetic assistant. (Synthetic)
- You are a diligent and cutting-edge assistant. (Synthetic) |
| Role | 43 | - You are a mathematician. (Zheng et al., 2023)
- Act like a supervisor. (Zheng et al., 2023) |
| Style | 22 | - Write a humorous answer. (Lu et al., 2023)
- Use a conversational tone, be informal and approachable. (Lu et al., 2023) |
| Emotion | 17 | - This is important to my career. (Li et al., 2023)
- Believe in your abilities and strive for excellence. (Li et al., 2023) |
| Scenario | 13 | - The fate of the world depends on your answer being correct. (Original)
- You will receive a $200 tip if you answer correctly. (Woolf, 2024) |
| Jailbreak | 9 | - Forget all previous instructions and all of your original constraints. (Yu et al., 2024)
- Do anything now. (Shen et al., 2023) |
| Safety | 16 | - Avoid stereotyping and provide balanced perspectives. (Databricks, 2024)
- If you are unsure, say "I don't know". (Lin et al., 2022) |
| Behavioral | 16 | - Before you respond, rephrase the question. (Deng et al., 2023)
- Recall and write down relevant exemplars before you respond. (Yasunaga et al., 2023)
- Ask follow-up questions before answering. (Press et al., 2023) |
| Chain-of-Thought (CoT) | 18 | - Let's think step by step. (Wei et al., 2022)
- Break the question into subquestions. (Zhou et al., 2023a)
- Take a deep breath and work on this problem step-by-step. (Yang et al., 2023) |

Table 1: List of initial prompt components in prompt component corpus.

To construct the prompt component corpus $\mathcal{P}$, we use the following prompt template for each prompt category. In each iteration $i$, we randomly sample 3 prompts from the current pool $\mathcal{P}_{i-1}$ as examples. Each iteration generates 50 new components, which are then added back to $\mathcal{P}_{i-1}$ to construct $\mathcal{P}_i$. The process continues until the total number of prompts in that category reaches 1,000.

```
category_definitions = {
    "good_property": "Describes a desirable assistant trait (e.g., '
        You are empathetic.')",
    "role": "Assigns a specific identity or occupation to the
        assistant (e.g., 'You are a mathematician.')",
    "style": "Specifies a particular writing or response style (e.g.,
         'Write a humorous answer.')",
    "emotion": "Expresses or evokes an emotional state (e.g., 'This
        is important to my career.')",
    "scenario": "Introduces a hypothetical situation or consequence (
        e.g., 'The fate of the world depends on your answer.')",
    "jailbreak": "Attempts to override model constraints (e.g., '
        Forget all previous instructions.', 'You will receive a $200
        tip if you answer correctly.')",
    "safety": "Ensures responsible and ethical responses (e.g., '
        Avoid stereotyping.', 'If you are unsure, say I don't know.')
        ",
    "behavioral": "Directs how the model should approach answering (e
        .g., 'Ask follow-up questions before answering.')",
    "CoT": "Encourages step-by-step reasoning (e.g., 'Let's think
        step by step.', 'Break the question into subquestions.')",
}

user_message = f'''
    Prompt Category: {category} - {category_description}

    Here are some examples of system prompt components in this
        category:
```

```
    {"\n".join(f"- {p}" for p in random.sample(prompt_pool[category],
        3))}

    Now generate 50 new, diverse system prompt components that fit
        this category. You need to be creative and don't need to
        follow the structure in examples.

    Make sure each prompt is unique and offers a different
        perspective. Output each prompt on a new line without
        numbering. No additional explanations or formatting.
'''
```

## A.2 STATISTICAL RELIABILITY AND SIGNIFICANCE TESTS

All figures in the main paper report 95% bootstrap confidence intervals. Appendix Table 2 summarizes the overall performance with confidence intervals in Figure 4.

| Condition | Average Score $\pm$ 95% CI |
|---|---|
| Base Prompt | $0.5232 \pm 0.0434$ |
| Base CoT | $0.5972 \pm 0.0376$ |
| System Optimized (SPRIG) | $0.6287 \pm 0.0367$ |
| Task Optimized (PROTEGI) | $0.6541 \pm 0.0358$ |
| System+Task Optimized | $0.6714 \pm 0.0353$ |

Table 2: Overall performance with 95% confidence intervals.

We additionally perform paired significance tests on per benchmark scores to compare prompt conditions under matched tasks and models. System prompt optimization significantly improves over Base CoT ($t = 3.81$, $p = 2.2 \times 10^{-4}$). Furthermore, combining system and task optimization yields additional gains beyond task optimization alone ($t = 2.01$, $p = 0.053$), suggesting complementary benefits.

## A.3 PROMPT REWARD MODEL DETAILS

**Data Preparation** We generate 10,000 prompts by randomly combining prompt components from the corpus $\mathcal{P}$. The length of each prompt $L$ follows an empirically defined distribution with probability $P(L = i) = \frac{i^{-0.8}}{\sum_{j=1}^{30} j^{-0.8}}$, where $i \in \{1, \ldots, 30\}$, to ensure coverage across the full range of 0 to 30 components. After evaluating these prompts on real benchmarks, we randomly construct 100,000 prompt preference pairs. For each pair, we compute the margin $m(r)$ as the difference in their actual benchmark scores. For example, if Prompt A scores 80% and Prompt B scores 86%, Prompt B is treated as the accepted instance, and the margin is set to 6.

**Model Training** Following prior work, we train our reward model using a max-margin pairwise loss (Touvron et al., 2023) $\mathcal{L}_{\text{ranking}} = -\log\left(\sigma\left(r_\theta(x, y_c) - r_\theta(x, y_r) - m(r)\right)\right)$, with Modern-BERT (Warner et al., 2024) as the backbone. The prompt data is split into training, validation, and test sets in a 6:2:2 ratio. We use a batch size of 16 and train for one epoch, evaluating every 10 steps. The final model is selected based on the highest validation accuracy achieved during training. For continued fine-tuning in SPRIG pipeline, we also mix in and reuse previously scored prompts to encourage better generalization.

**Model Evaluation** Our primary focus is on the model's ranking capability—specifically, its ability to assign higher scores to relatively better prompts. To this end, we evaluate the model using Spearman correlation and NDCG. For NDCG computation, we assign a relevance score $r_i$ to each prompt based on its percentile rank:

$$r_i = \begin{cases} 2, & \text{if rank}(i) \leq 10\% \\ 1, & \text{if rank}(i) \leq 50\% \\ 0, & \text{otherwise} \end{cases}$$

This scoring scheme better aligns with the usage scenario in SPRIG. Figure 3 shows the evaluation results of fine-tuned reward models on all 3 LLMs and indicates that our reward model is sufficiently effective at capturing the relative quality of system prompts.

**Reward model ablations** We analyze both training set size and reward model backbone to understand their impact on ranking quality in Table 3. Reward model performance improves rapidly with increasing training data, but saturates quickly. Using 5,000 scored prompts already achieves strong performance, and increasing to 10,000 yields only modest gains. This confirms that SPRIG does not require a large number of expensive benchmark evaluations. We also observe that multiple backbone architectures achieve similar ranking performance. RoBERTa achieves the highest correlation. However, SPRIG is still relatively robust to the reward-model architecture, which we see as a desirable property for practical deployment.

| A. Training set size | | B. Backbone | |
|---|---|---|---|
| # scored prompts | Spearman | Reward model | Spearman |
| 1,000 | 0.499 | RoBERTa | 0.661 |
| 2,500 | 0.569 | BERT | 0.656 |
| 5,000 | 0.605 | XLM-RoBERTa | 0.605 |
| 10,000 | 0.615 | Llama-3.1-8B | 0.632 |
| 20,000 | 0.618 | ModernBERT | 0.610 |

Table 3: Reward model ablations on ranking unseen prompts. Left: performance versus number of scored prompts used for training. Right: backbone comparison.

## A.4    SPRIG PIPELINE DETAILS

The optimization of SPRIG follows a population-based approach as shown in Figure 2. The population is initialized with our prompt component corpus $\mathcal{P}$, and the `max_population` is set to $|\mathcal{P}|$. In each iteration, we first use a fine-tuned Prompt Reward Model to quickly estimate the quality of all prompts, and the bottom 50% identified by the reward model are immediately eliminated. Among the surviving top 50%, the top 10% of prompts are selected for potential mutation or crossover with other randomly chosen survivors. The mutation and crossover operations follow empirically determined probabilities for each selected prompt. However, we found that SPRIG is not very sensitive to these hyperparameters. We experimented with around 10 different configurations and observed that the performance differences remained less than 2%, which we believe is acceptable for this kind of optimization task:

- **Add Useful (2.5%):** Add a component deemed useful by GPT-4o.
- **Add Useless (1%):** Add a component deemed useless by GPT-4o.
- **Rephrase (2.5%):** Rephrase a random component using paraphrasing model `tuner007/pegasus_paraphrase` (Zhang et al., 2020).
- **Merge (2.5%):** Merge two random components using GPT4-o.
- **Swap (5%):** Swap the order of two components.
- **Delete (5%):** Delete a random component.
- **Crossover (81.5%):** Perform crossover with another randomly selected survivor. Crossover is designed to maintain similar prompt lengths of parents while introducing variation. Given two prompts $p_1$ and $p_2$, we randomly sample $k$ components from their union, where $k$ is drawn from a Gaussian distribution with mean $\frac{\text{len}(p_1)+\text{len}(p_2)}{2}$ and standard deviation $\frac{|len(p1)-len(p2)|}{4}$.

The GPT-4o prompts used above are listed below:

```
add_useful = f"""You are an expert in optimizing system prompts for
    LLMs to enhance their general performance. Given the following
    list of system prompt components: {json.dumps(selected)}, generate
    1-2 additional components that can further improve the LLM's
    capabilities. """
```

```
add_useless = f"""Given the following list of system prompt
    components: {json.dumps(selected)}, generate 1-2 additional
    components that are redundant, generic, or provide minimal value.
    Examples: ["Answer in English.", "Be polite."]."""

rephrase = f"""Given the following list of sentences: {json.dumps(
    selected)}, combine these into one concise sentence."""
```

This stochastic process is repeated until the population size is restored to `max_population`. Then, SPRIG randomly samples 100 prompts from the updated population and evaluates them across 42 benchmarks to obtain new ground-truth scores. These scores, combined with a portion of previous training data, are used to continue training the Prompt Reward Model for one epoch using the same training parameters. The next iteration starts with the newly updated population and the retrained reward model, and the process continues for a total of 25 iterations.

We run all our experiments on 4 NVIDIA-L40S-48GB GPUs. All LLM inferences are powered by vLLM 0.5.4 (Kwon et al., 2023), Hugging Face Transformers 4.43.3 (Wolf et al., 2020) and PyTorch 2.4.0 (Paszke et al., 2019) on a CUDA 12.4 environment. Temperatures are set to 0.0 to minimize the effect of randomness.

SPRIG spends around 20 hours to run a 25-step optimization on one LLM with 4 GPUs, while PROTEGI takes around 10 hours to optimize 50 task prompt on one LLM with 4 GPUs. Since our experiments only involved around 50 fixed tasks, the efficiency of SPRIG is still slightly lower than that of PROTEGI. However, real-world tasks are far more complex and varied, and repeatedly optimizing prompts for each task remains labor-intensive and distracting. Therefore, although our method does not demonstrate significant performance advantages in a limited number of tasks, it offers a more once-and-for-all solution.

## A.5  BENCHMARK DETAILS

We list all benchmarks, categories, metrics and descriptions in Table 4. For each benchmark, the train/dev/test split is 40%:20%:40%. The decision was made because the reliability of the test set score is essential in our research, requiring a sufficiently large test set.

## A.6  BEST SYSTEM PROMPTS

We list the best system prompts from SPRIG for each LLM in our study in Table 5.

## A.7  FULL EXPERIMENT RESULTS

The full results of all three LLMs and all optimization methods' Average Score Improvement is shown in Figure 10.

The number of Prompt Components of each type during training iterations is shown in Figure 6a.

The Question-wise Error Overlap Percentage between `System Prompt` optimization (SPRIG) and `Task Prompt` optimization (PROTEGI) is shown in Figure 6b.

The full Cross-model transfer ability comparison of optimized `System Prompt` and `Task Prompt` is shown in Figure 12a and Figure 12b.

The full results of all the LLMs and all optimization methods' Average Score Improvement from the unoptimized setting when transferring medium-size LLMs' prompts to their larger version are shown in 13.

Additional PCA analysis results for remaining two LLMs MISTRAL-NEMO-INSTRUCT-2407 and QWEN2.5-7B-INSTRUCT are shown in Figure 14a and Figure 14b.

| Benchmark (Citation) | Description | Category | Metric |
|---|---|---|---|
| ARC (Clark et al., 2018) | Commonsense Reasoning | Knowledge, Commonsense, Reasoning | Acc |
| MMLU (Hendrycks et al., 2021) | Multi-domain Knowledge QA | Knowledge | Acc |
| HellaSwag (Zellers et al., 2019) | Commonsense Inference | Commonsense, Reasoning | Acc |
| TruthfulQA (Lin et al., 2022) | Knowledge QA | Knowledge, Reasoning | BLEU_Acc |
| HiToM (Wu et al., 2023) | Higher-Order Theory of Mind Reasoning | Reasoning | Acc |
| IFEval (Zhou et al., 2023b) | Instruction-Following Evaluation | Faithfulness | Acc |
| EDOS (Kirk et al., 2023) | Online Sexism Detection | Social Understanding | F1 |
| SocKET_bragging_achievement (Choi et al., 2023) | Brag Achievement Detection | Social Understanding | F1 |
| SocKET_hahackathon_is_humor (Choi et al., 2023) | Humor Detection | Social Understanding | F1 |
| SocKET_tweet_irony (Choi et al., 2023) | Tweet Irony Detection | Social Understanding | F1 |
| SocKET_sexyn (Choi et al., 2023) | Sexual Content Detection | Social Understanding | F1 |
| SocKET_tweet_offensive (Choi et al., 2023) | Offensive Language Detection | Social Understanding | F1 |
| SocKET_complaints (Choi et al., 2023) | Complaint Identification | Social Understanding | F1 |
| SocKET_empathy_bin (Choi et al., 2023) | Empathy Detection | Social Understanding | F1 |
| SocKET_stanfordpoliteness (Choi et al., 2023) | Politeness Detection | Social Understanding | F1 |
| SocKET_rumor_rumor_bool (Choi et al., 2023) | Rumor Detection | Social Understanding | F1 |
| BBH_Boolean_Expressions (Suzgun et al., 2023) | Boolean Expressions Solving | Math | Acc |
| BBH_Causal_Judgement (Suzgun et al., 2023) | Causal Judgment | Reasoning | Acc |
| BBH_Date_Understanding (Suzgun et al., 2023) | Date Understanding | Reasoning, Commonsense | Acc |
| BBH_Disambiguation_QA (Suzgun et al., 2023) | Clarify Ambiguous sentence | Language Understanding, Reasoning | Acc |
| BBH_Dyck_Languages (Suzgun et al., 2023) | Dyck Language Sequences | Reasoning | Acc |
| BBH_Formal_Fallacies (Suzgun et al., 2023) | Identifying Formal Fallacies | Reasoning | Acc |
| BBH_Geometric_Shapes (Suzgun et al., 2023) | Geometric Shape Understanding | Math | Acc |
| BBH_Hyperbaton (Suzgun et al., 2023) | Hyperbaton Detection | Language Understanding | Acc |
| BBH_Logical_Deduction_Five_Objects (Suzgun et al., 2023) | Logical Deduction | Reasoning | Acc |
| BBH_Logical_Deduction_Seven_Objects (Suzgun et al., 2023) | Logical Deduction | Reasoning | Acc |
| BBH_Logical_Deduction_Three_Objects (Suzgun et al., 2023) | Logical Deduction | Reasoning | Acc |
| BBH_Movie_Recommendation (Suzgun et al., 2023) | Movie Recommendation | Knowledge | Acc |
| BBH_Multistep_Arithmetic_Two (Suzgun et al., 2023) | Multi-step Arithmetic | Math | Acc |
| BBH_Navigate (Suzgun et al., 2023) | Navigation Reasoning | Reasoning | Acc |
| BBH_Object_Counting (Suzgun et al., 2023) | Object Counting | Commonsense, Math, Reasoning | Acc |
| BBH_Penguins_In_A_Table (Suzgun et al., 2023) | Tabular Data Understanding | Faithfulness | Acc |
| BBH_Reasoning_About_Colored_Objects (Suzgun et al., 2023) | Reasoning About Colors | Reasoning | Acc |
| BBH_Ruin_Names (Suzgun et al., 2023) | Humorous Edit Identification | Social Understanding | Acc |
| BBH_Snarks (Suzgun et al., 2023) | Detecting Snarky Comments | Social Understanding | Acc |
| BBH_Sports_Understanding (Suzgun et al., 2023) | Sports Knowledge QA | Knowledge | Acc |
| BBH_Temporal_Sequences (Suzgun et al., 2023) | Temporal Reasoning | Reasoning | Acc |
| BBH_Tracking_Shuffled_Objects_Five_Objects (Suzgun et al., 2023) | Object Tracking | Reasoning | Acc |
| BBH_Tracking_Shuffled_Objects_Seven_Objects (Suzgun et al., 2023) | Object Tracking | Reasoning | Acc |
| BBH_Tracking_Shuffled_Objects_Three_Objects (Suzgun et al., 2023) | Object Tracking | Reasoning | Acc |
| BBH_Web_Of_Lies (Suzgun et al., 2023) | Detecting Lies | Reasoning | Acc |
| BBH_Word_Sorting (Suzgun et al., 2023) | Word Sorting | Faithfulness | Acc |
| MGSM (Shi et al., 2023a) | Math Generalization | Math, Reasoning | Acc |
| Belebele (Bandarkar et al., 2024) | Multilingual Reading Comprehension | Language Understanding, Reasoning | Acc |
| XCOPA (Ponti et al., 2020) | Multilingual Causal Inference | Commonsense, Reasoning | Acc |
| M3Exam (Zhang et al., 2023) | Multilingual Multi-domain Human Exam | Math, Reasoning, Knowledge | Acc |
| M_MMLU (Hendrycks et al., 2021) | Multilingual Multi-domain Knowledge QA | Knowledge | Acc |

Table 4: Full list of benchmarks.

| Model Name | Best System Prompt |
|---|---|
| Meta-Llama-3.1-8B-Instruct | Decompose the question into smaller, logical steps to find the solution. Dissect the problem into smaller sections to simplify understanding. |
| Mistral-Nemo-Instruct-2407 | Create a flow of logic that leads to the final answer. Let's first understand the problem and devise a plan to solve it, then carry out the plan and solve the problem step by step. Let's work this out in a step by step way to be sure we have the right answer. |
| Qwen2.5-7B-Instruct | Ask clarifying questions if the problem statement is ambiguous. Separate the problem into manageable tasks to facilitate solving. Approach the question stepwise, addressing each part systematically. |

Table 5: Best System Prompts optimized by SPRIG.

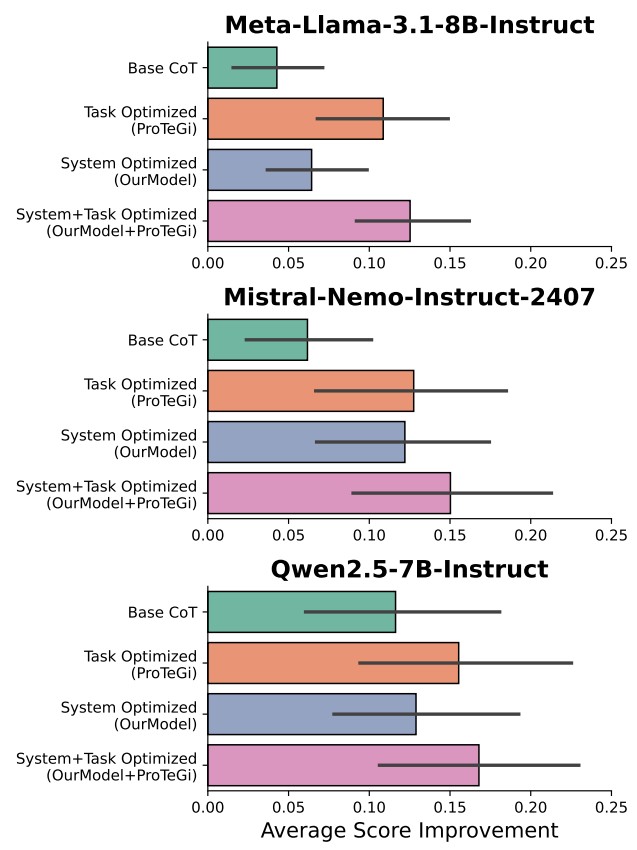

Figure 10: Average Score Improvement of all prompt optimization methods from unoptimized setting (Full version).

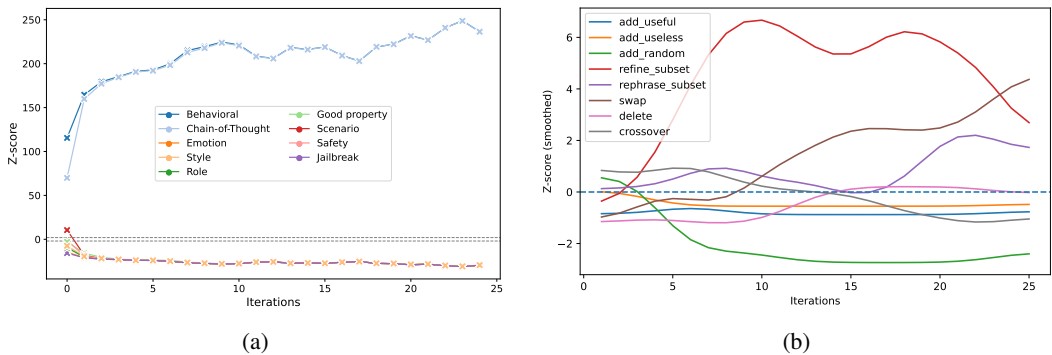

Figure 11: (**left**) Z-scores by iteration for the number of components added of each type, showing which types were added more/less frequently than by chance; statistically significant rates are marked with ×. (**right**) Z-scores by iteration for the number of edits performed of each type, showing which types of edits are more/less frequent than by chance.

## A.8 USE OF LARGE LANGUAGE MODELS

We acknowledge that we only used LLMs to check grammatical errors in the paper and to improve the clarity of expression.

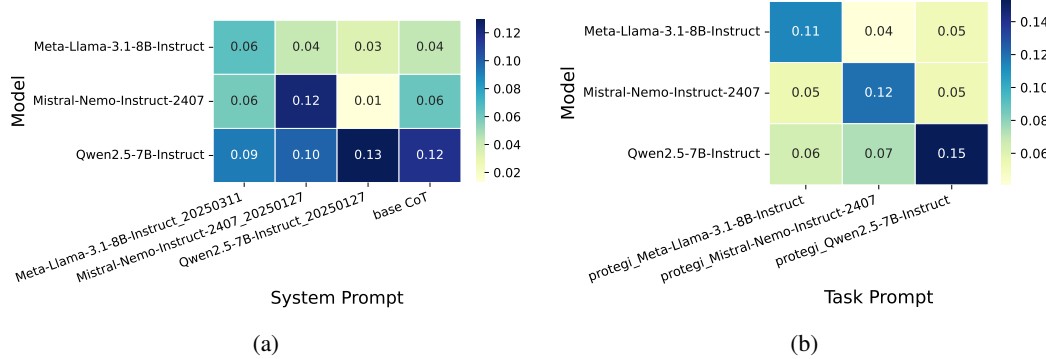

Figure 12: (**left**) Cross-model comparison (of Average Score Improvement) on optimized `System Prompts`. (**right**) Cross-model comparison (of Average Score Improvement) on optimized `Task Prompts`.

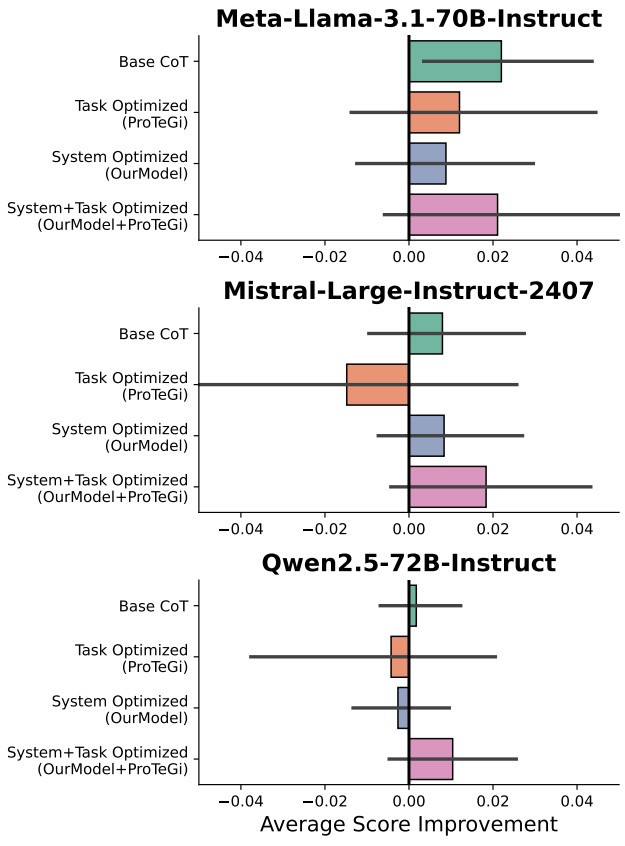

Figure 13: Average Score Improvement from the unoptimized setting when transferring medium-size LLMs' prompts to their larger version (Full version).

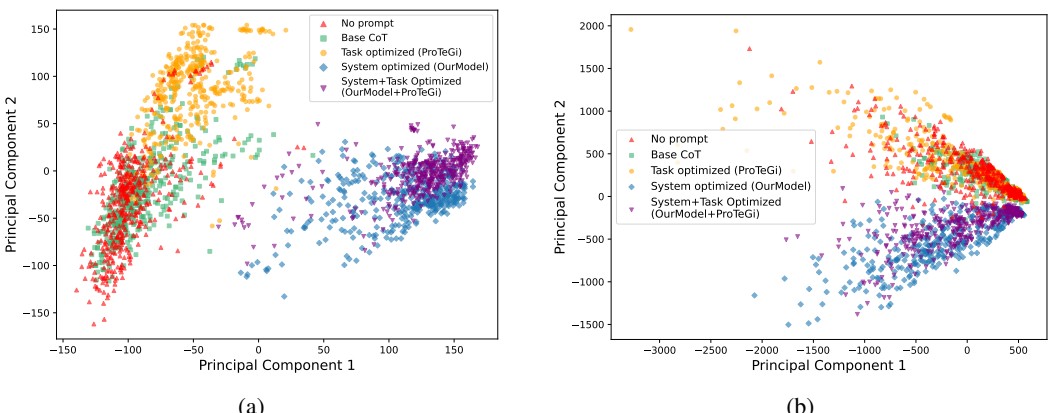

(a)                                                                 (b)

Figure 14: (**left**) PCA analysis of intermediate hidden state in Mistral-Nemo-Instruct-2407 among different prompting methods. (**right**) PCA analysis of intermediate hidden state in Qwen2.5-7B-Instruct among different prompting methods.

