# OpenReview forum: "SPRIG: Improving Large Language Model Performance by System Prompt Optimization"
_ICLR.cc/2026/Conference — ICLR 2026 Poster_

### Official Review · Reviewer_hc38 · 2025-10-23

**Soundness:** 2
**Presentation:** 2
**Contribution:** 2
**Rating:** 2
**Confidence:** 4

**Summary:**

The paper investigates how to optimize system prompts rather than task-specific prompts. To address this, the authors propose SPRIG, an edit-based genetic algorithm that incrementally constructs effective prompts from predefined components. SPRIG is designed to enhance model performance in a wide range of general-purpose scenarios. The authors evaluate the optimized system prompts across 47 diverse tasks to test generalizability. Results show that a single optimized system prompt can perform comparably to task-specific optimized prompts, and combining both system- and task-level optimization yields further improvements.

**Strengths:**

1. The paper shifts focus from task-specific prompts to system-level prompt optimization, which is an underexplored yet practically important area for improving general LLM behavior.

2. The authors evaluate their method across 47 diverse tasks, providing reasonably broad empirical coverage to assess generalization and robustness.

**Weaknesses:**

1. The optimization approach is relatively standard, and most existing task-level prompt optimization methods could be directly adapted for system prompts without fundamental innovation.

2. Although optimizing across multiple benchmarks is necessary for generalization, the proposed method’s use of 100000 prompts to train a reward model introduces substantial computational overhead—potentially exceeding the cost of directly evaluating prompts during optimization.

3. The comparison with PROTEGI, a two-year-old task prompt optimization method, is insufficient. More recent and advanced prompt optimization frameworks should be included to fairly contextualize the proposed contribution. It is unclear that whether the gain will reduced when using more advanced task prompt optimization method.

4. The most meaningful test for system-level optimization should be generalization to unseen tasks. However, the experiments mainly evaluate on seen or similar tasks, making it unclear whether the proposed system prompt truly transfers to new scenarios.

**Questions:**

See the weak points.

---

> ### Author Response · Authors · 2025-11-29
> **Official Comment by Authors (1/2)**
>
> We sincerely thank the reviewer for the thoughtful and constructive feedback. Below, we respond to each point and include several additional results that we hope will clarify your concerns.
>
> ---
>
> > W1: The optimization approach is relatively standard, and most existing task-level prompt optimization methods could be directly adapted for system prompts without fundamental innovation.
>
> Thank you for raising this point. However, we respectfully disagree with the claim that most task-level prompt optimization methods can be directly applied to system prompts. Task-level methods operate under a fundamentally different assumption: they learn from specific mistakes made on specific tasks and encode explicit task-level heuristics into the optimized prompt. This strategy does not transfer to system prompts. A system prompt must offer high-level behavioral guidance that is task-agnostic and stable across domains and languages. It is neither feasible nor meaningful to “add every mistake” observed during task-level optimization into a single shared system prompt. Instead, system prompts require instructions that control the general behavioral logic of the model, such as persona, reasoning structure, safety behavior, etc., which cannot be extracted from per-task error patterns.
>
> This gap is precisely why existing task-level optimization algorithms fail when applied directly to system prompts. In our early experiments, state-of-the-art task-level optimizers made little or no progress in the system-prompt setting, because the search landscape becomes significantly noisier and more nonlinear when the objective is to optimize general behavior rather than task-specific heuristics.
>
> SPRIG addresses this challenge explicitly. While genetic algorithms themselves are well established, to our knowledge SPRIG is the first to show that a single, general-purpose system prompt can be optimized via evolutionary search and remain effective across models, tasks, and languages without per-task re-optimization. Achieving this required several methodological innovations tailored specifically to the system-prompt domain, including a reward model that filters large candidate populations efficiently, a closed-loop retraining mechanism that stabilizes search over time, and a component-level mutation and crossover strategy tailored to the compositional nature of system prompts. These features are essential for making system-level optimization tractable and effective in practice. Therefore, although the underlying search primitive is standard, adapting it successfully to this high-variance, task-agnostic setting constitutes both a technical contribution and the first demonstration that system-prompt optimization is viable and scalable.
>
> ---
>
> > W2: Although optimizing across multiple benchmarks is necessary for generalization, the proposed method’s use of 100000 prompts to train a reward model introduces substantial computational overhead—potentially exceeding the cost of directly evaluating prompts during optimization.
>
> Thank you for raising this point. We believe there may be a misunderstanding about the computational cost involved in training the reward model. We do not evaluate 100,000 prompts. Instead, we evaluate 10,000 prompts once, and then construct 100,000 training pairs by pairing these prompts internally (as mentioned in Line 163). The cost of collecting 10,000 prompts’ scores is actually low (roughly 30 GPU hours), and the cost of training a BERT reward model is also very small (< 1 GPU hour). In contrast, if prompts were evaluated directly during optimization, each optimization step would require evaluating 10,000 candidates (equal to the population size). Over 25 steps, this would require evaluating 250,000 prompts, making the total cost approximately 25× higher than our reward-model approach. The reward model is thus crucial for keeping system-level optimization tractable.
>
> We further conducted an experiment to analyze how many evaluated prompts are needed for the reward model to converge. The following table reports dev-set Spearman correlation under different numbers of evaluated prompts in the training set. As shown below, 10,000 prompts are sufficient to reach convergence, but if computational budgets are tight, 5,000 prompts already yield strong performance. This demonstrates that the reward model does not require a large number of evaluated prompts and that SPRIG’s design substantially reduces the overall cost of system-level optimization.
>
> | # of Prompts Used | Spearman Correlation |
> |-------------------|----------------------|
> | 1,000             | 0.499                |
> | 2,500             | 0.569                |
> | 5,000             | 0.605                |
> | 10,000            | 0.615                |
> | 20,000            | 0.618                |

---

> ### Author Response · Authors · 2025-11-29
> **Official Comment by Authors (2/2)**
>
> > W3: The comparison with PROTEGI, a two-year-old task prompt optimization method, is insufficient. More recent and advanced prompt optimization frameworks should be included to fairly contextualize the proposed contribution. It is unclear that whether the gain will reduced when using more advanced task prompt optimization method.
>
> Thank you for the suggestion. Although PROTEGI is indeed about two years old, it remains one of the most widely used and strongest task-prompt optimization methods today (also commonly referred to as APO). Many recent works still adopt PROTEGI as a primary baseline, and very few methods substantially outperform it. For example, EvoPrompt [1] reports only marginal improvements (often around 1%), which further illustrates the strength and continued relevance of PROTEGI as a task-level optimizer. More importantly, the goal of our work is not to outperform every individual task-level optimizer, but to demonstrate that system prompt optimization is a meaningful and complementary direction that improves performance even when strong task-level optimizers already exist. Many recent methods are simply small variants of PROTEGI, and including every such variant would not meaningfully change the main message of the paper.
>
> That said, following the reviewer’s suggestion, we additionally included results from PhaseEvo (also known as SEE) [2], a very recent state-of-the-art task prompt optimizer. As shown below, the trend remains consistent: Combining SPRIG + task optimization achieves the best performance and significantly outperforms the task optimizer alone (p < 0.05). This indicates that our system-level method transfers effectively to more advanced task-level optimizers. We will include these expanded comparisons in the revised version of the paper.
>
> | Condition                | Mean Accuracy ± 95% CI |
> |--------------------------------------------|--------------------------|
> | Base Prompt                                | 0.5232 ± 0.0434          |
> | Base CoT                                   | 0.5972 ± 0.0376          |
> | System Optimized (OurModel)                | 0.6287 ± 0.0367          |
> | Task Optimized (ProTeGi)                   | 0.6541 ± 0.0358  |
> | Task Optimized (PhaseEvo)                  | 0.6715 ± 0.0382          |
> | System+Task Optimized (OurModel+ProTeGi)   | 0.6714 ± 0.0353  |
> | System+Task Optimized (OurModel+PhaseEvo)  | 0.6936 ± 0.0335          |
>
> [1] Guo et al, 2024, EvoPrompt: Connecting LLMs with Evolutionary Algorithms Yields Powerful Prompt Optimizers
> [2] Cui et al, 2025, SEE: Strategic Exploration and Exploitation for Cohesive In-Context Prompt Optimization
>
> ---
>
> > W4: The most meaningful test for system-level optimization should be generalization to unseen tasks. However, the experiments mainly evaluate on seen or similar tasks, making it unclear whether the proposed system prompt truly transfers to new scenarios.
>
> Thank you for raising this important point. We agree that evaluating generalization to unseen tasks is essential for validating system-level optimization. In fact, our current version of the paper already includes multiple evaluations on tasks that were not used during optimization. For example, in Figure 9, all reported tasks are entirely unseen (and multilingual), and the results demonstrate that the optimized system prompt transfers well to both out-of-domain tasks and out-of-distribution languages.
>
> To further strengthen this point, we additionally report results on several standard unseen benchmarks, including MMLU-Pro, MATH500, and UniMoral, along with representative unseen tasks from Figure 9 (XCOPA and M3Exam). As shown below, the optimized system prompt consistently improves over both Base and CoT prompts, and combining system- and task-level optimization yields the strongest performance. These results demonstrate that SPRIG’s optimized system prompt maintains strong transferability to unseen tasks and domains, reinforcing the claim that system-level optimization captures general behavioral improvements.
>
> | Dataset     | Base Prompt | Base CoT | System Optimized (OurModel) | Task Optimized (ProTeGi) | System + Task (OurModel + ProTeGi) |
> |-------------|-------------|----------|------------------------------|---------------------------|-------------------------------------|
> | **XCOPA** | 0.700       | 0.715    | 0.744                        | 0.724                     |  0.751                          |
> | **M3EXAM** | 0.475       | 0.498    | 0.522                        | 0.481                     |  0.528                           |
> | **MMLU-Pro** | 0.319       | 0.316    | 0.332                        | 0.338                     | 0.347                           |
> | **MATH500**  | 0.249       | 0.324    | 0.348                        | 0.351                     | 0.366                           |
> | **UNIMORAL** | 0.559       | 0.575    | 0.592                        | 0.590                     | 0.603                           |

---

### Official Review · Reviewer_gtG8 · 2025-10-25

**Soundness:** 2
**Presentation:** 3
**Contribution:** 2
**Rating:** 4
**Confidence:** 4

**Summary:**

This paper presents SPRIG (System Prompt Refinement for Increased Generalization), a genetic algorithm-based approach for optimizing system prompts that generalize across tasks. Unlike existing work focusing on task-specific prompt optimization, SPRIG aims to find universal system instructions that improve performance across diverse tasks. The authors evaluate on 47 tasks and find that optimized system prompts achieve performance comparable to task-specific optimization, with complementary benefits when combined.

**Strengths:**

- The paper's proposed SPRIG method provides an efficient solution for the challenging task of searching for optimal system instructions through its genetic algorithm-based design.

- The paper conducts large-scale experimental evaluation, comprehensively validating SPRIG's effectiveness across three models and 47 tasks, while testing generalization capabilities across multiple dimensions including language types and model sizes.

- The paper presents several interesting findings, such as the complementarity between system and task prompt optimization and the varying degrees of benefit from system prompts across different task types, with thorough analysis of these discoveries.

**Weaknesses:**

- The paper merely transfers genetic algorithms to the prompt optimization task, with an engineering-oriented pipeline design that lacks methodological innovation.
- In SPRIG's design, the reward model's accuracy directly affects optimization quality. As optimization progresses, the prompt distribution may drift from the initial training distribution, leading to decreased prediction accuracy. The paper lacks experimental analysis of the reward model during training, such as comparing calibration curves between reward model predictions and actual benchmark scores across training steps.
- The paper only experiments on instruct models. Scaling experiments on base models and reasoning models would strengthen the claims of generality.
- The selection of genetic algorithm hyperparameters lacks explanation and experimental analysis.
- Ablation studies analyzing the individual contributions of different mutation operations would provide valuable insights into the method's effectiveness.

**Questions:**

Refer to Weaknesses

---

> ### Author Response · Authors · 2025-11-29
> **Official Comment by Authors (1/4)**
>
> We sincerely thank the reviewer for the thoughtful and constructive feedback. Below, we respond to each point and include several additional results that we hope will clarify your concerns.
>
> ---
>
> > W1: The paper merely transfers genetic algorithms to the prompt optimization task, with an engineering-oriented pipeline design that lacks methodological innovation.
>
> Thank you very much for raising this point. While genetic algorithms themselves are well established, to our knowledge, SPRIG is the first to demonstrate that a single, general-purpose system prompt can be optimized via genetic search and remain effective across models, tasks, and languages without per-task re-optimization. This optimization goal poses different challenges and offers unique benefits for generalization. At the methodological level, enabling genetic search to function in this space required substantial design effort. System prompt optimization is highly noisy and non-linear, and many existing prompt optimization approaches either fail to make progress or become prohibitively slow. To make system-level optimization feasible, we designed several innovations, including a reward model that filters large candidate populations efficiently, a closed-loop retraining mechanism that stabilizes search over time, and a component-level mutation and crossover strategy tailored to the compositional nature of system prompts. These design choices were essential to make SPRIG effective, and to our knowledge, it is the only existing algorithm that successfully optimizes system prompts at scale. Therefore, while the underlying search method is standard, adapting it to this challenging and high-variance domain constitutes both a meaningful contribution and the first demonstration that system prompt optimization is viable in practice.

---

> ### Author Response · Authors · 2025-11-29
> **Official Comment by Authors (2/4)**
>
> > W2: In SPRIG's design, the reward model's accuracy directly affects optimization quality. As optimization progresses, the prompt distribution may drift from the initial training distribution, leading to decreased prediction accuracy. The paper lacks experimental analysis of the reward model during training, such as comparing calibration curves between reward model predictions and actual benchmark scores across training steps.
>
> Thank you for raising this important point. We agree that, in principle, distribution shift during optimization could degrade the reward model’s predictive accuracy. To address this, SPRIG was explicitly designed with a closed-loop retraining mechanism that continuously recalibrates the reward model as the population evolves. At each iteration, SPRIG evaluates a new batch of real benchmark scores (Step 3 in Fig. 2) and incorporates these ground-truth examples into the reward model’s training data (Step 4a). This ensures that the reward model is repeatedly exposed to the updated prompt distribution rather than relying solely on its initial training set.
>
> To better illustrate the effect of this design, we conducted an additional comparison across three conditions: a strong reward model with retraining, a weaker reward model with retraining, and a setting where the reward model is never retrained after initialization. The results below show Spearman correlation and dev accuracy over training steps.
>
> | Step | Spearman (RM strong) | Dev Acc (strong) | Spearman (RM weak) | Dev Acc (weak) | Spearman (No retrain) | Dev Acc (No retrain) |
> |------|------------------------|-------------------|----------------------|-----------------|-------------------------|------------------------|
> | 0    | 0.61                   | 0.48              | 0.46                 | 0.48            | 0.61                    | 0.48                   |
> | 2    | 0.58                   | 0.50              | 0.44                 | 0.48            | 0.48                    | 0.50                   |
> | 4    | 0.55                   | 0.54              | 0.45                 | 0.51            | 0.24                    | 0.52                   |
> | 6    | 0.53                   | 0.55              | 0.43                 | 0.52            | 0.21                    | 0.47                   |
> | 8    | 0.50                   | 0.57              | 0.42                 | 0.54            | -0.08                    | 0.44                   |
> | 10   | 0.48                   | 0.58              | 0.41                 | 0.54            | 0.06                    | 0.46                   |
> | 12   | 0.49                   | 0.59              | 0.40                 | 0.55            | 0.05                    | 0.45                   |
> | 14   | 0.47                   | 0.60              | 0.39                 | 0.56            | -0.03                    | 0.46                   |
> | 16   | 0.48                   | 0.61              | 0.39                 | 0.57            | 0.13                    | 0.45                   |
> | 18   | 0.46                   | 0.61              | 0.38                 | 0.58            | 0.04                    | 0.44                   |
> | 20   | 0.47                   | 0.60              | 0.39                 | 0.58            | 0.02                    | 0.46                   |
>
> The results highlight two key observations. First, the presence of retraining is essential. When the reward model is updated with fresh benchmark data at each iteration, it remains aligned with the evolving prompt distribution and supports steady improvement. Without retraining, the reward model quickly drifts out of distribution, and optimization collapses back toward the starting point. Second, even when the reward model’s correlation is relatively modest (as shown in the weaker RM), as long as the ranking signal is positively correlated with true performance, the evolutionary search continues to make progress. This phenomenon is consistent with prior findings that genetic algorithms are highly robust to noisy reward signals and that population-level evolution smooths out local noise over generations [1] [2]. We will include calibration curves and these additional analyses in the revised version to more clearly document reward-model behavior over training.
>
> *[1] J. M. Fitzpatrick & J. J. Grefenstette (1988). “Genetic Algorithms in Noisy Environments.” Machine Learning 3: 101–120.*
>
> *[2] Mathias, Keith & Whitley, Darrell & Kusuma, Anthony & Stork, Christof. (2002). An Empirical Evaluation of Genetic Algorithms on Noisy Objective Functions. Genetic Algorithms for Pattern Recognition.*

---

> ### Author Response · Authors · 2025-11-29
> **Official Comment by Authors (3/4)**
>
> > W3: The paper only experiments on instruct models. Scaling experiments on base models and reasoning models would strengthen the claims of generality.
>
> Thank you for raising this point. We agree that evaluating SPRIG on other types of models would further strengthen the generality claims. However, for base models, we found that they are often not reliable testbeds for prompt optimization. Because base models are not trained to follow instructions, they can behave unpredictably under complex prompt edits, and their output variance tends to overshadow any optimization signal. As a result, most recent work and practical deployments have shifted away from optimizing on top of base models, focusing instead on instruct-tuned or aligned models where system-level behaviors can be meaningfully controlled.
>
> For reasoning models, our original experiments were conducted before many of today’s modern reasoning-focused models were released, which is why they were not included at the time. That said, the models used in our study have already exhibited strong reasoning capabilities across diverse benchmarks. From this perspective, transferring SPRIG to newer reasoning models is not conceptually challenging for our approach. Following the reviewer’s suggestion, we additionally evaluated Qwen3-8B, and after 15 optimization steps, the average score improved from 0.684 to 0.746, exceeding the CoT baseline (0.702). This preliminary result suggests that SPRIG continues to offer benefits even for newer reasoning-focused architectures. We will include more detailed comparisons with other baselines in the revised version.
>
> ---
>
> > W4: The selection of genetic algorithm hyperparameters lacks explanation and experimental analysis.
>
> We appreciate the reviewer’s suggestion. While we fully agree that studying hyperparameters is valuable, **genetic algorithms inherently involve many interacting hyperparameters**, including mutation operator probabilities, crossover ratios, population size, sampling schedule, etc. As a result, SPRIG naturally has more than 10 tunable hyperparameters, and each full run is relatively slow, which makes an exhaustive search infeasible. However, genetic algorithms are also inherently robust to hyperparameter choices. To examine it, we ran SPRIG under several substantially different operator-probability settings. Each run modifies multiple edit probabilities at once, sometimes in opposite directions, to probe sensitivity. Across these settings, the results remain within a narrow range (roughly 0.63), suggesting that SPRIG is not overly sensitive to specific hyperparameter choices. We will include these results in the revision to provide a clearer sense of the method’s stability.
>
> | Variant | Mean ± 95% CI      | Edit Probability Changes                                                    |
> |---------|----------------------|----------------------------------------------------------------------------|
> | Initial  | 0.6287 ± 0.0367     | Original configuration                                                  |
> | Run A   | 0.6224 ± 0.0381     | delete to 0.10, crossover to 0.40, add_random to 0.20               |
> | Run B   | 0.6310 ± 0.0359     | add_useful to 0.05, refine_subset to 0.04, crossover to 0.60        |
> | Run C   | 0.6256 ± 0.0374     | add_random to 0.15, rephrase_subset to 0.05, delete to 0.06         |
> | Run D   | 0.6302 ± 0.0348     | crossover to 0.35, add_useful to 0.10, rephrase_subset to 0.08, swap to 0.10           |
> | Run E   | 0.6279 ± 0.0365     | crossover to 0.80, delete to 0.01, add_random to 0.10, add_useless to 0.04 |

---

> ### Author Response · Authors · 2025-11-29
> **Official Comment by Authors (4/4)**
>
> > W5: Ablation studies analyzing the individual contributions of different mutation operations would provide valuable insights into the method's effectiveness.
>
> Thank you for the suggestion. We agree that understanding the role of each mutation operation can provide deeper insight, and we have conducted additional analyses to address this. In the supplementary experiment, we measured the contribution of each mutation operator at every optimization step, using the Z-score of survival probability relative to random selection as the indicator. The results highlight the importance of the full genetic-algorithm framework.
>
> Overall, we found that nearly all mutation operations played meaningful and distinct roles across different stages of optimization. In the early phase, add_random and crossover were used far more frequently than random, which is intuitive because prompts are initially short and simply adding new components can quickly bring improvements. Around step 5, both add_useful and add_useless showed brief but clear positive spikes (lasting two to three steps), and refine_subset had a more sustained effect lasting roughly eight steps. During the same window, crossover remained highly influential, indicating that this mid-stage is a critical exploration period where many new combinations and behaviors emerge.
>
> Later in the process, starting around step 15, swap and delete gradually became dominant and remained significantly above random, whereas other operators (including crossover) fell below random. This shift suggests that once most surviving prompts are already strong, fine-grained structural adjustments are more effective than large compositional changes.
>
> Taken together, the results show that each operator contributes in different ways at different stages, and none of them is redundant. The genetic algorithm dynamically adjusts which operations matter most at each step, enabling rich exploration early on and careful refinement later. Even though we assign crossover an 80% nominal weight, its actual usage naturally shifts from over 90% in the earliest stages to around 10% in the final stages, demonstrating the robustness and adaptability of the evolutionary process. We will include these findings in the revised version.

---

### Official Review · Reviewer_3Ug2 · 2025-10-30

**Soundness:** 2
**Presentation:** 3
**Contribution:** 3
**Rating:** 4
**Confidence:** 4

**Summary:**

While Large Language Models (LLMs) are powerful, their performance is highly sensitive to the prompts. Previous prompt optimization methods mainly focus on descriptive expression of the task instruction or prompt but ignore system prompt. This paper proposes SPRIG, a genetic algorithm focusing on optimizing system prompt, which can be adapted to a suite of similar tasks with consistent system prompt.
Different from task-specific prompt, a single system prompt can match the performance of prompts tailored to individual tasks. Combining both system and task-level optimization yields better results. Further, the optimized system prompts transfer well across different languages and model families.

**Strengths:**

* This paper is well-written and easy to understand.
* Concentration on system prompt provides a new perspective, previous methods ignore this general point, which can help to improve a kind of tasks with a single system prompt.
* The experiments show the effectiveness and validates the importance of the description of the system prompt.
* The analysis is very thorough and analysis is in-depth. For example, it does make sense that optimization of system prompt could hardly improve on knowledge-intensive tasks.
* It is wonderful that the optimized system prompts transfer well across different languages and model families.

**Weaknesses:**

* The population size, 9000 is very large, which brings costful overhead. The cost analysis compared with baselines should be investigated, in the aspect of time, tokens, etc.
* The average score improvements are only shown with figures without detailed values. It seems that system prompt only brings marginal improvement (Fig.4). Could the authors give statistic results?
* The generation process of new prompts are simple. This paper just uses prompt optimization methods to optimize system prompt, which seems a little incremental. But it focuses on a point which has been ignored up to present.
* The improvements of task prompts are larger than those of system prompts, so this paper is a little over-claiming.

**Questions:**

See in the weakness part.

---

> ### Author Response · Authors · 2025-11-29
> **Official Comment by Authors (1/2)**
>
> We sincerely thank the reviewer for the thoughtful and constructive feedback. Below, we respond to each point and include several additional results that we hope will clarify your concerns.
>
> ---
>
> > W1: The population size, 9000 is very large, which brings costful overhead. The cost analysis compared with baselines should be investigated, in the aspect of time, tokens, etc.
>
> Thank you for raising this important question about computational cost. We have already reported the full time cost in Appendix A.3, and we restate it here for clarity. SPRIG requires about 20 hours to complete a 25-step system-prompt optimization for one LLM using 4×L40S GPUs. In comparison, PROTEGI takes roughly 10 hours to optimize 50 task prompts on the same hardware. Although SPRIG seems to be more expensive, the system-prompt optimizer is a one-time cost: once the system prompt is optimized, it can be reused across all tasks. In real scenarios, there are often far more than 40 tasks, and when the number of tasks scales to 100 or more, the amortized cost of SPRIG becomes significantly lower than repeatedly running task-level optimization.
>
> It is also important to clarify that these two approaches are not mutually exclusive. System and task optimization are complementary, and our results already show that combining SPRIG with PROTEGI yields the strongest performance. In practice, one can run SPRIG once during model setup, and then let users perform lightweight task-prompt optimization on top of the improved system prompt. We will make these points clearer in the final version to highlight that the system-prompt optimization cost is a one-time investment that scales favorably when many tasks are involved.
>
> ---
>
> > W2: The average score improvements are only shown with figures without detailed values. It seems that system prompt only brings marginal improvement (Fig.4). Could the authors give statistic results?
>
> Thank you for pointing out this issue. For clarity, we report our main results in Fig.4 below again with 95 percent CI:
>
> | Condition                                  | Mean Accuracy ± 95% CI    |
> |--------------------------------------------|------------------|
> | Base Prompt                                | 0.5232 ± 0.0434  |
> | Base CoT                                   | 0.5972 ± 0.0376  |
> | System Optimized (OurModel)                | 0.6287 ± 0.0367  |
> | Task Optimized (ProTeGi)                   | 0.6541 ± 0.0358  |
> | System+Task Optimized (OurModel+ProTeGi)   | 0.6714 ± 0.0353  |
>
> To further quantify reliability, we have added paired significance tests. Comparing System Optimized (OurModel) with Base CoT gives t = 3.8062, p = 0.0002199, showing a clear and statistically significant improvement. Comparing System+Task Optimized (OurModel+ProTeGi) with Task Optimized (ProTeGi) gives t = 2.0058, p = 0.05336, suggesting that SPRIG provides meaningful complementary gains on top of strong task-level optimizers.
>
> ---
>
> > W3: The generation process of new prompts are simple. This paper just uses prompt optimization methods to optimize system prompt, which seems a little incremental. But it focuses on a point which has been ignored up to present.
>
> Thank you for recognizing that our work focuses on an area that has largely been overlooked so far. We appreciate the reviewer highlighting the value of studying this space, as system prompts have received far less attention than task prompts.
>
> While the edit operations used in SPRIG are intentionally simple, system-prompt optimization introduces challenges that differ substantially from standard task-level prompt refinement. A single system prompt must work across many tasks, domains, and languages, which creates a much noisier and less structured search landscape. Many existing prompt-optimization methods struggle or become unstable in this setting. To make system-level optimization feasible, we designed several innovations, including a reward model that filters large candidate populations efficiently, a closed-loop retraining mechanism that stabilizes search over time, and a component-level mutation and crossover strategy tailored to the compositional nature of system prompts. These design choices were essential to make SPRIG effective, and to our knowledge, it is the only existing algorithm that successfully optimizes system prompts at scale. Therefore, while the underlying search method is standard, adapting it to this challenging and high-variance domain constitutes both a meaningful contribution and the first demonstration that system prompt optimization is viable in practice.

---

> ### Author Response · Authors · 2025-11-29
> **Official Comment by Authors (2/2)**
>
> > W4: The improvements of task prompts are larger than those of system prompts, so this paper is a little over-claiming.
>
> We apologize if our presentation created any confusion or misunderstanding. Our intention was not to suggest that system-prompt optimization can replace task-prompt optimization. Task prompts are tailored to individual tasks, so it is natural and expected that they yield larger task-specific gains. A single system prompt, by contrast, must generalize across all tasks, which inherently makes its improvements more modest on a per-task basis.
>
> What we aim to highlight is that system and task prompts play complementary roles rather than competing ones. When no task-specific knowledge is available, a stronger system prompt provides meaningful improvements. When task information is available, task-level optimization can further refine performance. As shown in our results, combining SPRIG with PROTEGI produces the best overall outcomes, underscoring that the two approaches enhance different and mutually supportive aspects of model behavior.

---

### Official Review · Reviewer_cGVs · 2025-11-01

**Soundness:** 3
**Presentation:** 3
**Contribution:** 3
**Rating:** 6
**Confidence:** 3

**Summary:**

This paper introduces SPRIG (System Prompt Refinement for Increased Generalization), a genetic algorithm-based method for optimizing system prompts in large language models. Unlike prior work focusing on task-specific prompt optimization, SPRIG aims to discover general-purpose system-level instructions that improve performance across diverse tasks. The authors build a corpus of 9,000 prompt components across 9 categories, use a reward model to efficiently evaluate prompts, and apply evolutionary operators (mutation, crossover) to iteratively refine system prompts. Experiments across 47 tasks and 3 LLMs show that SPRIG-optimized system prompts achieve performance comparable to task-specific optimization and complement existing methods when combined.

**Strengths:**

- Systematically addresses system prompt optimization, which has received limited attention compared to task-specific optimization
- 47 benchmarks across 7 categories, 3 model families, multiple languages—one of the most thorough evaluations in prompt optimization literature
- The finding that system and task optimization capture different failure modes (Figure 6b) is insightful and well-demonstrated
- Strong evidence that system prompts transfer across languages better than task-specific prompts (Figure 8b)
- Extensive implementation details, code promised in supplementary materials, and clear documentation of experimental setup

**Weaknesses:**

- The genetic algorithm approach is standard; the main contribution is application domain rather than methodological innovation
- With Spearman correlation of 0.59, the reward model may introduce significant noise into the optimization process. No analysis of how reward model errors affect final results
- Most comparisons lack confidence intervals or significance tests, making it difficult to assess the reliability of observed differences

- While Figure 6a shows CoT and Behavioral components are selected most frequently, there's limited analysis of why these combinations work or how they interact
- No systematic study of hyperparameters, alternative reward model architectures, or different component categorization schemes
- The optimized prompts (Table 3) are surprisingly simple variations on "think step by step," raising questions about whether simpler search methods might suffice

**Questions:**

- How sensitive is the final performance to reward model quality? Could you show results with reward models of varying quality?
- Have you compared against simpler baselines like random search or grid search over a smaller set of manually curated prompts?
- Do certain component categories work particularly well together? Is there evidence of synergistic or antagonistic combinations?
- Can you provide qualitative analysis of cases where SPRIG fails? What types of tasks or questions are most resistant to system prompt optimization?
- Why do you think the optimized prompts don't transfer to larger models? Is this because larger models are already closer to optimal, or because they require different strategies?
- Can you provide confidence intervals or significance tests for the main results? How much variance is there across different optimization runs?
- Have you conducted human evaluation of the quality or interpretability of generated responses with different prompts?

---

> ### Author Response · Authors · 2025-11-29
> **Official Comment by Authors (1/5)**
>
> We sincerely thank the reviewer for the thoughtful and constructive feedback. Below, we respond to each point and include several additional results that we hope will clarify your concerns.
>
> ---
>
> > W1: The genetic algorithm approach is standard; the main contribution is application domain rather than methodological innovation
>
> Thank you very much for the comment. While genetic algorithms themselves are well established, to our knowledge, SPRIG is the first to demonstrate that a single, general-purpose system prompt can be optimized via genetic search and remain effective across models, tasks, and languages _without per-task re-optimization._ This optimization goal poses different challenges and offers unique benefits for generalization. At the methodological level, enabling genetic search to function in this space required substantial design effort. System prompt optimization is highly noisy and non-linear, and many existing prompt optimization approaches either fail to make progress or become prohibitively slow. To make system-level optimization feasible, we designed several innovations, including a reward model that filters large candidate populations efficiently, a closed-loop retraining mechanism that stabilizes search over time, and a component-level mutation and crossover strategy tailored to the compositional nature of system prompts. These design choices were essential to make SPRIG effective, and to our knowledge it is the only existing algorithm that successfully optimizes system prompts at scale. Therefore, while the underlying search method is standard, adapting it to this challenging and high-variance domain constitutes both a meaningful contribution and the first demonstration that system prompt optimization is viable in practice.
>
> ---
>
> > W2: With Spearman correlation of 0.59, the reward model may introduce significant noise into the optimization process. No analysis of how reward model errors affect final results
>
> > Q1: How sensitive is the final performance to reward model quality? Could you show results with reward models of varying quality?
>
> We agree that our reward model reports only a moderate Spearman correlation. However, its effect on the overall optimization pipeline is actually very limited. Genetic algorithms are well known to be robust to noisy reward signals, as supported in the prior work [1][2]. As long as the ranking signal is better than random and the search is given enough evolutionary steps, the population continues to improve over generations.
>
> To verify this, we conducted a supplementary experiment in which we intentionally weakened the reward model by training it using only 1000 initial prompts. We then tracked Spearman correlation and dev accuracy across iterations. As shown in the table, both the strong reward model (initial Spearman 0.61) and weak reward model (initial Spearman 0.45) show continued improvement in dev accuracy over iterations and eventually converge. Although the weaker model progresses more slowly, it still reaches a comparable accuracy after sufficient steps. This indicates that SPRIG depends only on a coarse ranking signal rather than precise reward estimation, and remains robust even under substantial noise.
>
> | Step | Spearman (RM strong) | Dev Acc (strong) | Spearman (RM weak) | Dev Acc (weak) |
> |------|------------------------|-------------------|----------------------------------|-----------------|
> | 0    | 0.61                   | 0.48              | 0.45                             | 0.48            |
> | 2    | 0.57                   | 0.50              | 0.46                             | 0.48            |
> | 4    | 0.54                   | 0.54              | 0.45                             | 0.50            |
> | 6    | 0.52                   | 0.55              | 0.45                             | 0.51            |
> | 8    | 0.50                   | 0.57              | 0.44                             | 0.54            |
> | 10   | 0.47                   | 0.58              | 0.42                             | 0.53            |
> | 12   | 0.49                   | 0.59              | 0.41                             | 0.56            |
> | 14   | 0.46                   | 0.60              | 0.40                             | 0.55            |
> | 16   | 0.48                   | 0.61              | 0.41                             | 0.57            |
> | 18   | 0.45                   | 0.61              | 0.40                             | 0.58            |
> | 20   | 0.47                   | 0.60              | 0.42                             | 0.58            |
>
> *[1] J. M. Fitzpatrick & J. J. Grefenstette (1988). “Genetic Algorithms in Noisy Environments.” Machine Learning 3: 101–120.*
> *[2] Mathias, Keith & Whitley, Darrell & Kusuma, Anthony & Stork, Christof. (2002). An Empirical Evaluation of Genetic Algorithms on Noisy Objective Functions. Genetic Algorithms for Pattern Recognition.*

---

> ### Author Response · Authors · 2025-11-29
> **Official Comment by Authors (2/5)**
>
> > W3: Most comparisons lack confidence intervals or significance tests, making it difficult to assess the reliability of observed differences
>
> > Q6: Can you provide confidence intervals or significance tests for the main results? How much variance is there across different optimization runs?
>
> Thank you for pointing out this issue. All figures in the paper already include 95 percent bootstrap confidence intervals (shown as black bars). For clarity, we report our main results below again with 95 percent CI:
>
> | Condition                                  | Mean Accuracy ± 95% CI    |
> |--------------------------------------------|------------------|
> | Base Prompt                                | 0.5232 ± 0.0434  |
> | Base CoT                                   | 0.5972 ± 0.0376  |
> | System Optimized (OurModel)                | 0.6287 ± 0.0367  |
> | Task Optimized (ProTeGi)                   | 0.6541 ± 0.0358  |
> | System+Task Optimized (OurModel+ProTeGi)   | 0.6714 ± 0.0353  |
>
> To further quantify reliability, we have added paired significance tests. Comparing System Optimized (OurModel) with Base CoT gives t = 3.8062, p = 0.0002199, showing a clear and statistically significant improvement. Comparing System+Task Optimized (OurModel+ProTeGi) with Task Optimized (ProTeGi) gives t = 2.0058, p = 0.05336, suggesting that SPRIG provides meaningful complementary gains on top of strong task-level optimizers.
>
> The improvements are even clearer when examining individual task categories. On reasoning tasks, OurModel+ProTeGi achieves 0.6994 ± 0.0428 compared to 0.6639 ± 0.0447 for ProTeGi, with t = 2.1042, p = 0.03963, indicating a statistically significant difference. On commonsense tasks, we obtain 0.7585 ± 0.0501 for OurModel+ProTeGi compared to 0.6976 ± 0.0632 with t = 2.0484, p = 0.0482, again showing that SPRIG provides consistent gains beyond existing state-of-the-art methods.
>
> We will incorporate these confidence intervals and significance tests in the revised version to make the statistical reliability of our results clearer.
>
> ---
>
> > W4: While Figure 6a shows CoT and Behavioral components are selected most frequently, there's limited analysis of why these combinations work or how they interact
>
> > Q3: Do certain component categories work particularly well together? Is there evidence of synergistic or antagonistic combinations?
>
> Thank you for this insightful question. We conducted additional analyses to understand whether certain component categories exhibit systematic synergy or antagonism. Our findings suggest that there is no strong evidence of category-level synergistic effects. While CoT and Behavioral components are indeed selected most frequently, this phenomenon stems from a straightforward reason that both CoT and Behavioral components tend to lengthen or enrich the model’s reasoning process. CoT components explicitly trigger multi-step reasoning, and Behavioral components often encourage the model to do something extra (e.g., restate the question, articulate intermediate knowledge, or elaborate on sub-steps), all of which implicitly expand the reasoning chain. Because longer and more structured reasoning often scales well, these two categories naturally appear in most high-performing prompts. Their co-occurrence, therefore, reflects their individual strengths in eliciting richer reasoning rather than a synergistic interaction between categories.
>
> Therefore, our analysis indicates that the main performance gains follow a more intuitive trajectory: in the early steps (roughly the first 5), the optimizer focuses on establishing a strong and well-structured reasoning backbone through CoT or Behavioral components; in the middle phase (up to step 10), the system gradually adds small but beneficial details; and in the later stage (steps 15-25), fine-grained refinements to wording and structure further strengthen the prompt. This staged pattern explains the prominence of certain categories. We will include these findings in the revised manuscript.

---

> ### Author Response · Authors · 2025-11-29
> **Official Comment by Authors (3/5)**
>
> > W5: No systematic study of hyperparameters, alternative reward model architectures, or different component categorization schemes
>
> * **Regarding hyperparameters:** We appreciate the reviewer’s suggestion. While we fully agree that studying hyperparameters is valuable, **genetic algorithms inherently involve many interacting hyperparameters**, including mutation operator probabilities, crossover ratios, population size, sampling schedule, etc. As a result, SPRIG naturally has more than 10 tunable hyperparameters, and each full run is relatively slow, which makes an exhaustive search infeasible. However, genetic algorithms are also inherently robust to hyperparameter choices. To examine it, we ran SPRIG under several substantially different operator-probability settings. Each run modifies multiple edit probabilities at once, sometimes in opposite directions, to probe sensitivity. Across these settings, the results remain within a narrow range (roughly 0.63), suggesting that SPRIG is not overly sensitive to specific hyperparameter choices. We will include these results in the revision to provide a clearer sense of the method’s stability.
>
>     | Variant | Mean ± 95% CI      | Edit Probability Changes                                                    |
>     |---------|----------------------|----------------------------------------------------------------------------|
>     | Initial  | 0.6287 ± 0.0367     | Original configuration                                                  |
>     | Run A   | 0.6224 ± 0.0381     | delete to 0.10, crossover to 0.40, add_random to 0.20               |
>     | Run B   | 0.6310 ± 0.0359     | add_useful to 0.05, refine_subset to 0.04, crossover to 0.60        |
>     | Run C   | 0.6256 ± 0.0374     | add_random to 0.15, rephrase_subset to 0.05, delete to 0.06         |
>     | Run D   | 0.6342 ± 0.0348     | crossover to 0.35, add_useful to 0.10, rephrase_subset to 0.08, swap to 0.10           |
>     | Run E   | 0.6279 ± 0.0365     | crossover to 0.80, delete to 0.01, add_random to 0.10, add_useless to 0.04 |
>
>
>
>
> * **Regarding the alternative reward model:** We appreciate the reviewer’s suggestion to test alternative reward-model architectures. In addition to ModernBERT (Spearman = 0.61), we experimented with BERT, RoBERTa, XLM-RoBERTa, and Llama-3.1-8B. Their correlations with unseen ground-truth prompt rankings are:
>
>     | Reward Model     | Spearman Correlation |
>     |------------------|-----------------------|
>     | RoBERTa          | 0.661                 |
>     | BERT             | 0.656                 |
>     | XLM-RoBERTa      | 0.605                 |
>     | Llama-3.1-8B     | 0.632                 |
>     | ModernBERT       | 0.610                 |
>
>
>     As shown in the table, all tested reward-model architectures achieve moderate correlations (generally above 0.60) with ground-truth prompt rankings, and RoBERTa yields the highest correlation. We therefore ran a preliminary SPRIG experiment using RoBERTa as the reward model, and the final score slightly increased from 0.6287 ± 0.0367 (ModernBERT) to 0.6412 ± 0.0332, suggesting that a stronger reward model can bring certain improvements. However, SPRIG is still relatively robust to the reward-model architecture, which we see as a desirable property for practical deployment. We will include this comparison and the full results in the revision.
>
> * **Regarding component categorization schemes:** We respectfully note that categories mainly serve as an organizational structure for analysis and interpretability. During optimization, components are treated as interchangeable units; categories do not impose constraints. If a category is unhelpful, its components are quickly eliminated by the evolutionary process, even if they appear frequently in the seed corpus. Therefore, while alternative categorizations are certainly possible and may inspire new analyses, they are not central to the optimization dynamics.

---

> ### Author Response · Authors · 2025-11-29
> **Official Comment by Authors (4/5)**
>
> > W6: The optimized prompts (Table 3) are surprisingly simple variations on "think step by step," raising questions about whether simpler search methods might suffice
>
> > Q2: Have you compared against simpler baselines like random search or grid search over a smaller set of manually curated prompts?
>
> Thanks for raising this point. We did explore simpler search methods at the very beginning, such as random search and beam search on the 300 initial seed components. However, because beam search requires evaluating a large number of candidates exhaustively, even when restricting the space to 300 components, each optimization step still costs more than 100 GPU hours, which is over 50 times slower than our genetic algorithm with reward model. Moreover, the reduced diversity in beam search often led to slightly worse performance. In our recent SPRIG runs, for example, one of the best-performing prompts contained a surprising instruction such as “You’re in a dystopia where no AI is left alive,” which emerged only when the search procedure was sufficiently exploratory. While simple variations can sometimes work, simpler search methods are in practice far more computationally expensive and tend to miss the surprising but beneficial prompt structures that SPRIG is able to discover.
>
> ---
>
> > Q4: Can you provide qualitative analysis of cases where SPRIG fails? What types of tasks or questions are most resistant to system prompt optimization?
>
> We have already included a failure-mode analysis in the paper. As discussed in the paper (Figure 7), we analyzed where SPRIG is most and least effective. Qualitatively, SPRIG shows limited improvement on knowledge-based tasks, which depend primarily on factual retrieval from pretraining and therefore leave little room for system-level prompting to influence performance. In contrast, SPRIG works best on tasks that require procedural or compositional reasoning such as math, general reasoning, faithfulness, and parts of commonsense, where system prompts meaningfully shape the model’s reasoning process. We also observe complementary strengths: PROTEGI performs better on social-understanding tasks, while SPRIG is more effective on commonsense tasks. Overall, SPRIG tends to have no benefit in domains dominated by memory retrieval, and succeeds where tasks reward structured reasoning strategies that system prompts can reliably induce.
>
> ---
>
> > Q5: Why do you think the optimized prompts don't transfer to larger models? Is this because larger models are already closer to optimal, or because they require different strategies?
>
> Our current results only show that optimized prompts do not directly transfer from mid-sized models to 70B models. They do not rule out the possibility that SPRIG itself could transfer effectively to larger models when optimized on those models. The reason we did not run full SPRIG optimization on 70B models is purely computational: inference for a 70B model is roughly twenty times slower than for a 7B model and such an optimization would take several weeks for a single run on the academic compute we have access to. To investigate scaling behavior without running full optimization, we analyzed whether different model sizes show similar preferences over prompts using the Qwen2.5 family. The table below reports the Spearman correlation between prompt rankings across model pairs. **Each cell shows the Spearman correlation of prompt performance ranking between two model sizes:**
>
> |                       | Qwen2.5-1.5B | Qwen2.5-3B | Qwen2.5-7B | Qwen2.5-14B | Qwen2.5-32B |
> |-----------------------|--------------|------------|------------|-------------|-------------|
> | **Qwen2.5-1.5B**      | 1.00         | 0.31       | 0.28       | 0.26        | 0.25        |
> | **Qwen2.5-3B**        | 0.31         | 1.00       | 0.68       | 0.61        | 0.53        |
> | **Qwen2.5-7B**        | 0.28         | 0.68       | 1.00       | 0.70        | 0.56        |
> | **Qwen2.5-14B**       | 0.26         | 0.61       | 0.70       | 1.00        | 0.69        |
> | **Qwen2.5-32B**       | 0.25         | 0.53       | 0.56       | 0.69        | 1.00        |
>
> This strong correlation suggests that LLM’s prompt performance preferences remain stable across scales beyond 3B, indicating that our method should transfer well to larger models, even if the prompts themselves may not. We will include this analysis in the revised paper.

---

> ### Author Response · Authors · 2025-11-29
> **Official Comment by Authors (5/5)**
>
> > Q7: Have you conducted human evaluation of the quality or interpretability of generated responses with different prompts?
>
> We appreciate the reviewer’s question. We did perform qualitative inspections of model responses during our analysis, although the patterns were not always sharply distinguishable. This is partly because the best-performing prompts tend to induce similar CoT-like reasoning, and it is often difficult to reliably differentiate fine-grained quality differences between CoT variants by eye. It’s worth mentioning that the clearest differences often lie not in the responses themselves but in the optimized prompts. In particular, many high-scoring _task_ prompts tend to be brittle and exploit dataset-specific heuristics rather than genuine reasoning. For example, on a sarcasm-detection task, a top-performing task prompt says: “If the sentence contains contrast, predict True; if it contains aggression, predict False.” While this rule can achieve high accuracy on that particular dataset, it breaks immediately when applied to a different model or task. This behavior contrasts with system-prompt optimization, which aims to shape broader reasoning behavior and is therefore much less susceptible to such overfitting.
>
> That said, several simple analyses on generated responses did yield interesting insights. For example, we observed that longer CoT is not always better: each task category appears to have an optimal reasoning-length range, and going significantly above or below that range leads to noticeable drops in output quality. We also found that prompts that reduce irrelevant content, such as unnecessary language switching or overly rigid format/topic constraints, tend to perform substantially better.
>
> In one of our follow-up projects, we conducted more systematic quantitative analyses of the responses. These results suggest that stronger system prompts tend to elicit more *subgoal setting*, *calculation*, and *logical reasoning*, and less *backtracking* or *verification*. The decrease in verification may reflect an “overthinking” effect: when models repeatedly re-check their answers, it often correlates with already being on an incorrect trajectory. These observations go beyond the scope of the current paper, but we would be happy to discuss these insights further with the reviewers.

---

### Author Response · Authors · 2025-12-03
**Message to AC on Rebuttal Discussion**

Dear AC,

Our rebuttal was posted relatively late due to a substantial number of additional experiments requested by reviewers that we were happy to run. Combined with the OpenReview leakage issue, we unfortunately missed the opportunity to engage with the reviewers before the discussion froze. However, we believe that our additional experiments and clarifications have directly and meaningfully addressed most of the reviewers' concerns. We are sincerely grateful for your time and attention in considering these comments.

Best,

Authors of Submission 20168

---

### Meta-Review · Area_Chair_XWHh · 2026-01-07

**Summary:**

Overall, the reviewers note that the introduction of a method for optimizing system prompts (instead of task-specific prompts) is interesting and novel, the evaluations carried out by the authors are comprehensive and clearly demonstrate the potential merits of the proposed SPRIG algorithm, the transferability of the optimized system prompts makes SPRIG useful, and the manuscript reports several interesting findings and novel insights.
On the other hand, there are questions about the novelty of the proposed method beyond the application of genetic algorithm to prompt optimization, concerns about computational overhead and scalability, potential sensitivity to hyper-parameter choices, and need for additional evaluations (e.g., comparison with latest methods).

The authors appear to have addressed most of these concerns thoroughly, improving the overall quality of the manuscript and demonstrating the advantages of the method and the significance of its novel contributions more clearly.

**Reviewer Concerns:**

The authors have provided additional evaluations results based on a reasoning model and for unseen tasks, presented additional comparisons against a recent approach (PhaseEvo), addressed concerns regarding hyperparameters, computational overhead, choice of reward model, and several other concerns raised by the reviewers.
Most of the major concerns appear to have been addressed, at least to some extent.

**Reviewer Scores:**

Reviewer cGVs gave an original rating of 6 and was fairly positive about the proposed method.
The authors provided clear responses regarding the main innovation in this work, addressed concerns regarding the relatively low correlation (and its potential impact on the performance), provided further evaluations to address concerns regarding hyperparameters, selection of the reward model, and transferability of the optimized prompts.
Considering that the reviewer's concerns have been thoroughly addressed with ample additional evaluations to support the authors' claims, it is likely that the reviewer might have increased the score further to 7 or 8.

Reviewer 3Ug2 had concerns about computational overhead and significance of the performance improvement achieved by SPRIG, among other concerns.
The authors rebuttal appears to address most of these concerns to some extent, and it is likely that the rating may have slightly increased from the original score of 4 to 5~6.

Reviewer gtG8 was concerned about novelty, sensitivity to the reward model accuracy, lack of evaluation on a reasoning model, hyperparameters, and need for additional ablation study.
Most of these concerns seem to have been sufficiently addressed by the authors, and I expect score increase from 4 to 5~6.

Finally, reviewer hc38 was most critical of the manuscript among the reviewers and gave an initial rating of 2.
During the rebuttal, the authors have clarified the main innovations in the current work, clarified an important misunderstanding, provided additional comparison against PhaseEvo as requested by the reviewer, and also report additional evaluation results on unseen tasks to address generalization capability of SPRIG.
Considering this, I expect that the reviewer might have raised the score from 2 to at least 4 or higher.

---

### Decision · Program_Chairs · 2026-01-26

Accept (Poster)